# The Unbalanced Gromov Wasserstein Distance: Conic Formulation and Relaxation

## Abstract

Comparing metric measure spaces (i.e. a metric space endowed with a probability distribution) is at the heart of many machine learning problems. This includes for instance predicting properties of molecules in quantum chemistry or generating graphs with varying connectivity. The most popular distance between such metric measure spaces is the Gromov-Wasserstein (GW) distance, which is the solution of a quadratic assignment problem. This distance has been successfully applied to supervised learning and generative modeling, for applications as diverse as quantum chemistry or natural language processing. The GW distance is however limited to the comparison of metric measure spaces endowed with a *probability* distribution. This strong limitation is problematic for many applications in ML where there is no a priori natural normalization on the total mass of the data. Furthermore, imposing an exact conservation of mass across spaces is not robust to outliers and often leads to irregular matching. To alleviate these issues, we introduce two Unbalanced Gromov-Wasserstein formulations: a distance and a more tractable upper-bounding relaxation. They both allow the comparison of metric spaces equipped with arbitrary positive measures up to isometries. The first formulation is a positive and definite divergence based on a relaxation of the mass conservation constraint using a novel type of quadratically-homogeneous divergence. This divergence works hand in hand with the entropic regularization approach which is popular to solve large scale optimal transport problems. We show that the underlying non-convex optimization problem can be efficiently tackled using a highly parallelizable and GPU-friendly iterative scheme. The second formulation is a distance between mm-spaces up to isometries based on a conic lifting. Lastly, we provide numerical simulations to highlight the salient features of the unbalanced divergence and its potential applications in ML.

## 1 Introduction

Comparing data distributions on different metric spaces is a basic problem in machine learning. This class of problems is for instance at the heart of surfaces (Bronstein et al., 2006) or graph matching (Xu et al., 2019) (equipping the surface or graph with its associated geodesic distance), regression problems in quantum chemistry (Gilmer et al., 2017) (viewing the molecules as distributions of points in $\mathbb{R}^3$) and natural language processing (Grave et al., 2019; Alvarez-Melis & Jaakkola, 2018) (where texts in different languages are embedded as points distributions in different vector spaces).

**Metric measure spaces.** The mathematical way to formalize these problems is to model the data as *metric measure spaces* (mm-spaces). A mm-space is denoted as $\mathcal{X} = (X, d, \mu)$ where $X$ is a complete separable set endowed with a distance $d$ and a positive Borel measure $\mu \in \mathcal{M}_+(X)$. For instance, if $X = (x_i)_i$ is a finite set of points, then $\mu = \sum_i m_i \delta_{x_i}$ (here $\delta_{x_i}$ is the Dirac mass at $x_i$) is simply a set of positive weights $m_i = \mu(\{x_i\}) \geq 0$ associated to each point $x_i$, which accounts for its mass or importance. For instance, setting some $m_i$ to 0 is equivalent to removing the point $x_i$. We refer to Sturm (2012) for a mathematical account on the theory of mm-spaces.

In all the applications highlighted above, it makes sense to perform the comparisons up to isometric transformations of the data. Two mm-spaces $\mathcal{X} = (X, d_X, \mu)$ and $\mathcal{Y} = (Y, d_Y, \nu)$ are

considered to be equal (denoted $\mathcal{X} \sim \mathcal{Y}$) if they are isometric, meaning that there is a bijection $\psi : \mathrm{spt}(\mu) \rightarrow \mathrm{spt}(\nu)$ (where $\mathrm{spt}(\mu)$ is the support of $\mu$) such that $d_X(x, y) = d_Y(\psi(x), \psi(y))$ and $\psi_\sharp \mu = \nu$. Here $\psi_\sharp$ is the push-forward operator, so that $\psi_\sharp \mu = \nu$ is equivalent to imposing $\nu(A) = \mu(\psi^{-1}(A))$ for any set $A \subset Y$. For discrete spaces where $\mu = \sum_i m_i \delta_{x_i}$, then one should have $\nu = \psi_\sharp \mu = \sum_i m_i \delta_{\psi(x_i)}$. As highlighted by Mémoli (2011), considering mm-spaces up to isometry is a powerful way to formalize and analyze a wide variety of problems such as matching, regression and classification of distributions of points belonging to different spaces. The key to unlock all these problems is the computation of a distance between mm-spaces up to isometry. So far, existing distances (reviewed below) assume that $\mu$ is a probability distribution, i.e. $\mu(X) = 1$. This constraint is not natural and sometimes problematic for most of the practical applications to machine learning. The goal of this paper is to alleviate this restriction. We define for the first time a class of distances between unbalanced metric measure spaces, these distances being upper-bounded by divergences which can be approximated by an efficient numerical scheme.

**Csiszár divergences** The simplest case is when $X = Y$ and one simply ignores the underlying metric. One can then use Csiszár divergences (or $\varphi$-divergences), which perform a pointwise comparison (in contrast with optimal transport distances, which perform a displacement comparison). It is defined using an entropy function $\varphi : \mathbb{R}_+ \rightarrow [0, +\infty]$, which is a convex, lower semi-continuous, positive function with $\varphi(1) = 0$. The Csiszár $\varphi$-divergence reads $\mathrm{D}_\varphi(\mu|\nu) \overset{\text{def.}}{=} \int_X \varphi(\frac{\mathrm{d}\mu}{\mathrm{d}\nu}) \mathrm{d}\nu + \varphi'_\infty \int_X \mathrm{d}\mu^\perp$, where $\mu = \frac{\mathrm{d}\mu}{\mathrm{d}\nu}\nu + \mu^\perp$ is the Lebesgue decomposition of $\mu$ with respect to $\nu$ and $\varphi'_\infty = \lim_{r \rightarrow \infty} \varphi(r)/r \in \mathbb{R} \cup \{+\infty\}$ is called the recession constant. This divergence $\mathrm{D}_\varphi$ is convex, positive, 1-homogeneous and weak* lower-semicontinuous, see Liero et al. (2015) for details. Particular instances of $\varphi$-divergences are Kullback-Leibler (KL) for $\varphi(r) = r\log(r) - r + 1$ (note that $\varphi'_\infty = \infty$) and Total Variation (TV) for $\varphi(r) = |r - 1|$.

**Balanced and unbalanced optimal transport.** If the common embedding space $X$ is equipped with a distance $d(x, y)$, one can use more elaborated methods such as optimal transport (OT) distances, which are computed by solving convex optimization problems. This type of methods has proven useful for ML problems as diverse as domain adaptation (Courty et al., 2014), supervised learning over histograms (Frogner et al., 2015) and unsupervised learning of generative models (Arjovsky et al., 2017). In this case, the extension from probability distributions to arbitrary positive measures $(\mu, \nu) \in \mathcal{M}_+(X)^2$ is now well understood and corresponds to the theory of unbalanced OT. Following Liero et al. (2015); Chizat et al. (2018c), a family of unbalanced Wasserstein distances is defined by solving

$$\mathrm{UW}(\mu, \nu)^q \overset{\text{def.}}{=} \inf_{\pi \in \mathcal{M}(X \times X)} \int \lambda(d(x, y)) \mathrm{d}\pi(x, y) + \mathrm{D}_\varphi(\pi_1|\mu) + \mathrm{D}_\varphi(\pi_2|\mu). \quad (1)$$

Here $(\pi_1, \pi_2)$ are the two marginals of the joint distribution $\pi$, defined by $\pi_1(A) = \pi(A \times Y)$ for $A \subset X$. The mapping $\lambda : \mathbb{R}^+ \rightarrow \mathbb{R}$ and exponent $q \geq 1$ should be chosen wisely to ensure for instance that UW defines a distance (see Section 2.2.1). It is frequent to take $\rho \mathrm{D}_\varphi$ instead of $\mathrm{D}_\varphi$ (i.e. take $\psi = \rho\varphi$) to adjust the strength of the marginals' penalization. Balanced OT is retrieved with the convex indicator $\varphi = \iota_{\{1\}}$ or by taking the limit $\rho \rightarrow +\infty$, which enforces $\pi_1 = \mu$ and $\pi_2 = \nu$. When $0 < \rho < +\infty$, unbalanced OT operates a trade-off between transportation and creation of mass, which is crucial to be robust to outliers in the data and to cope with mass variations in the modes of the distributions. For supervised tasks, the value of $\rho$ should be cross-validated to obtain the best performances.

Its use is gaining popularity in applications, such as medical imaging registration (Feydy et al., 2019a), videos (Lee et al., 2019), generative learning (Balaji et al., 2020) and gradient flow to train neural networks (Chizat & Bach, 2018; Rotskoff et al., 2019). Furthermore, existing efficient algorithms for balanced OT extend to this unbalanced problem. In particular Sinkhorn's iterations, introduced in ML for balanced OT by Cuturi (2013), extend to unbalanced OT (Chizat et al., 2018a; Séjourné et al., 2019), as detailed in Section 3.

**The Gromov-Wasserstein distance and its applications.** The Gromov-Wasserstein (GW) distance (Mémoli, 2011; Sturm, 2012) generalizes the notion of OT to the setting of mm-spaces up to isometries. It corresponds to replacing the linear cost $\int \lambda(d)\mathrm{d}\pi$ of OT by a quadratic function

$$\mathrm{GW}(\mathcal{X}, \mathcal{Y})^q \overset{\text{def.}}{=} \min_{\pi \in \mathcal{M}_+(X \times Y)} \left\{ \int \lambda(|d_X(x, x') - d_Y(y, y')|) \mathrm{d}\pi(x, y) \mathrm{d}\pi(x', y') \; : \; \begin{matrix} \pi_1 = \mu \\ \pi_2 = \nu \end{matrix} \right\}, \quad (2)$$

It is proved in Mémoli (2011); Sturm (2012) that GW defines with $\lambda(t) = t^q$ a distance up to isometries on balanced mm-spaces (i.e. the measures are probability distributions). In this paper, we extend this construction to arbitrary positive measures, and provide explicit settings in Section 2.2.1.

This distance is applied successfully in natural language processing for unsupervised translation learning (Grave et al., 2019; Alvarez-Melis & Jaakkola, 2018), in generative learning for objects lying in spaces of different dimensions (Bunne et al., 2019) and to build VAE for graphs (Xu et al., 2020). It has been adapted for domain adaptation over different spaces (Redko et al., 2020). It is also a relevant distance to compute barycenters between graphs or shapes (Vayer et al., 2018; Chowdhury & Needham, 2020). When $(\mathcal{X}, \mathcal{Y})$ are Euclidean spaces, this distance compares distributions up to rigid isometry, and is closely related (but not equal) to metrics defined by procrustes analysis (Grave et al., 2019; Alvarez-Melis et al., 2019).

The problem (2) is non convex because the quadratic form $\int \lambda(|d_X - d_Y|)\mathrm{d}\pi \otimes \pi$ is not positive in general. It is in fact closely related to quadratic assignment problems (Burkard et al., 1998), which are used for graph matching problems, and are known to be NP-hard in general. Nevertheless, non-convex optimization methods have been shown to be successful in practice to use GW distances for ML problems. This includes for instance alternating minimization (Mémoli, 2011; Redko et al., 2020) and entropic regularization (Peyré et al., 2016; Gold & Rangarajan, 1996).

**Related works and contributions.** The concomitant work of De Ponti & Mondino (2020) extends the $L^p$ transportation distance defined in Sturm et al. (2006) to unbalanced mm-spaces and studies its geometric properties. This distortion distance is not equivalent to the GW distance, and is more difficult to estimate numerically because it explicitly imposes a triangle inequality constraint in the optimization problem. The work of Chapel et al. (2020) relaxes the GW distance to the unbalanced setting by hybridizing GW with partial OT (Figalli, 2010) for unsupervised labeling. It ressembles one particular setting of our formulation, but with some important differences, detailed in Section 2. Our construction is also connected to partial matching methods, which find numerous applications in graphics and vision (Cosmo et al., 2016). In particular, Rodola et al. (2012) introduces a mass conservation relaxation of the GW problem.

The two main contributions of this paper are the definition of two formulations relaxing the GW distance. The first one is called the Unbalanced Gromov-Wasserstein (UGW) divergence and can be computed efficiently on GPUs. The second one is called the Conic Gromov-Wasserstein distance (CGW). It is proved to be a distance between mm-spaces endowed with positive measures up to isometries, as stated in Theorem 1 which is the main theoretical result of this paper. We also prove in Theorem 1 that UGW can be used as a surrogate upper-bounding CGW. We present those concepts and their properties in Section 2. We also detail in Section 3 an efficient computational scheme for a particular setting of UGW. This method computes an approximate stationary point of the non-convex energy. It leverages the strength of entropic regularization and the Sinkhorn algorithm, namely that it is GPU-friendly and defines smooth loss functions amenable to back-propagation for ML applications. Section 4 provides some numerical experiments to highlight the qualitative behavior of this algorithm, which shed some lights on the favorable properties of UGW to cope with outliers and mass variations in the modes of the distributions.

## 2 UNBALANCED GROMOV-WASSERSTEIN FORMULATIONS

We present in this section our two new formulations and their properties. The first one, called UGW, is exploited in Sections 3 and 4 to derive an efficient algorithm used in numerical experiments. The second one, called CGW, defines a distance between mm-spaces up to isometries. Those results build upon the results of Liero et al. (2015), and a summary of the construction of UOT is detailed in Appendix A In all what follows, we consider complete separable mm-spaces endowed with a metric and a positive measure.

### 2.1 THE UNBALANCED GROMOV-WASSERSTEIN DIVERGENCE

This new formulation makes use of quadratic $\varphi$-divergences, defined as $\mathrm{D}_\varphi^\otimes(\rho|\nu) \stackrel{\text{def.}}{=} \mathrm{D}_\varphi(\rho \otimes \rho|\nu \otimes \nu)$, where $\rho \otimes \rho \in \mathcal{M}_+(X^2)$ is the tensor product measure defined by $\mathrm{d}(\rho \otimes \rho)(x, y) = \mathrm{d}\rho(x)\mathrm{d}\rho(y)$. Note that $\mathrm{D}_\varphi^\otimes$ is not a convex function in general.

**Definition 1** (Unbalanced GW). *The Unbalanced Gromov-Wasserstein divergence is defined as* $\mathrm{UGW}(\mathcal{X}, \mathcal{Y}) = \inf_{\pi \in \mathcal{M}^+(X \times Y)} \mathcal{L}(\pi)$ *where*

$$\mathcal{L}(\pi) \overset{\text{def.}}{=} \int_{X^2 \times Y^2} \lambda(|d_X(x, x') - d_Y(y, y')|) \mathrm{d}\pi(x, y) \mathrm{d}\pi(x', y') + \mathrm{D}_\varphi^\otimes(\pi_1|\mu) + \mathrm{D}_\varphi^\otimes(\pi_2|\nu). \quad (3)$$

This definition can be understood as an hybridation between (1) and (2) but with a twist: one needs to use the quadratic divergence $\mathrm{D}_\varphi^\otimes$ in place of $\mathrm{D}_\varphi$. To the best of our knowledge, it is the first time such quadratic divergences are being used and studied. In the TV case, this is the most important distinction between UGW and partial GW (Chapel et al., 2020). Note also that the balanced GW distance (2) is recovered as a particular case when using $\varphi = \iota_=$ or by letting $\rho \to +\infty$ for an entropy $\psi = \rho\varphi$.

Using quadratic divergences results in UGW being 2-homogeneous: for $\theta \geq 0$, writing $(\mathcal{X}_\theta, \mathcal{Y}_\theta)$ equiped with $(\theta\mu, \theta\nu)$, one has $\theta^{-2}\mathrm{UGW}(\mathcal{X}_\theta, \mathcal{Y}_\theta) = \mathrm{UGW}(\mathcal{X}, \mathcal{Y})$. When using non tensorized $\varphi$-divergences, the resulting unbalanced Gromov-Wasserstein functional between $\mathcal{X}_\theta$ and $\mathcal{Y}_\theta$ have very different and inconsistent behaviors when $\theta \to 0$ and $\theta \to +\infty$. Indeed, once normalized by $\theta^{-2}$ and $\theta^{-1}$, one obtains respectively balanced GW and a Hellinger-type distance. Using tensorized divergences ensure that the behavior does not depends on $\theta$.

We first prove the existence of optimal plans $\pi$ solution to (3), which holds for the three key settings of Section 2.2.1, namely for KL, TV, and for compact metric spaces (such as finite pointclouds and graphs). All proofs are deferred in Appendix B.

**Proposition 1** (Existence of minimizers). *We assume that $(X, Y)$ are compact and that either (i) $\varphi$ superlinear, i.e $\varphi'_\infty = \infty$, or (ii) $\lambda$ has compact sublevel sets in $\mathbb{R}_+$ and $2\varphi'_\infty + \inf \lambda > 0$. Then there exists $\pi \in \mathcal{M}_+(X \times Y)$ such that $\mathrm{UGW}(\mathcal{X}, \mathcal{Y}) = \mathcal{L}(\pi)$.*

The following proposition ensures that the functional UGW can be used to compare mm-spaces.

**Proposition 2** (Definiteness of UGW). *Assume that $\varphi^{-1}(\{0\}) = \{1\}$ and $\lambda^{-1}(\{0\}) = \{0\}$. Then $\mathrm{UGW}(\mathcal{X}, \mathcal{Y}) \geq 0$ and is 0 if and only if $\mathcal{X} \sim \mathcal{Y}$.*

We end this section with a reformulation of UGW (3) which is important to make the connection with the second formulation of the following section. Its proof is deferred to Appendix B. It splits UGW into two parts: the term $\varphi(0)(|(\mu \otimes \mu)^\perp| + |(\nu \otimes \nu)^\perp|)$ accounts for the pure creation/destruction of mass and a new transport cost $L_c$ accounts for the remaining part (partial/pure transport and partial creation/destruction of mass).

**Lemma 1.** *Defining $L_c(a, b) \overset{\text{def.}}{=} c + a\varphi(1/a) + b\varphi(1/b)$, and writing $(f \overset{\text{def.}}{=} \frac{\mathrm{d}\mu}{\mathrm{d}\pi_1}, g \overset{\text{def.}}{=} \frac{\mathrm{d}\nu}{\mathrm{d}\pi_2})$ the Lebesgue densities of $(\mu, \nu)$ w.r.t. $(\pi_1, \pi_2)$ such that $\mu = f\pi_1 + \mu^\perp$ and $\nu = g\pi_2 + \nu^\perp$, one has*

$$\mathcal{L}(\pi) = \int_{X^2 \times Y^2} L_{\lambda(|d_X - d_Y|)}(f \otimes f, g \otimes g) \mathrm{d}\pi \mathrm{d}\pi + \varphi(0)(|(\mu \otimes \mu)^\perp| + |(\nu \otimes \nu)^\perp|). \quad (4)$$

## 2.2 THE CONIC GROMOV-WASSERSTEIN DISTANCE

We introduce a second "conic" formulation of unbalanced GW, which is connected to UGW, and whose construction is inspired by the conic formulation of UOT (see Appendix A for an overview).

### 2.2.1 BACKGROUND ON CONE SETS AND DISTANCES

The conic formulation lifts a point $x \in X$ to a couple $(x, r) \in X \times \mathbb{R}^+$ where $r$ encodes some (power of a) mass. Then we seek optimal transport plans defined over $\mathfrak{C}[X] \overset{\text{def.}}{=} X \times \mathbb{R}_+/(X \times \{0\})$, where coordinates $(x, r = 0)$ with no mass are merged into a single point $\mathfrak{o}_X$ called the apex of the cone. In the sequel, points of $X \times \mathbb{R}_+$ are noted $(x, r)$, while $[x, r]$ are quotiented points of $\mathfrak{C}[X]$.

While transport plans depend on variables $([x, r], [y, s])$ and $([x', r'], [y', s'])$ in $\mathfrak{C}[X] \times \mathfrak{C}[Y]$, the transportation cost involved in our conic formulation only makes use of the 2-D cone $\mathfrak{C}[\mathbb{R}_+]$ over $\mathbb{R}_+$ endowed with the distance $|u - v|$ (note that any other distance on $\mathbb{R}$ could be used as well). More specifically, we consider coordinates of the form $([u, a], [v, b]) =$

$([d_X(x,x'), rr'], [d_Y(y,y'), ss']) \in \mathfrak{C}[\mathbb{R}_+] \times \mathfrak{C}[\mathbb{R}_+]$. Thus we now describe conic discrepancies $\mathcal{D}$ on $\mathfrak{C}[\mathbb{R}_+]$, which are defined for $(p,q) \geq 0$ as

$$\mathcal{D}([u,a],[v,b])^q \overset{\text{def.}}{=} H_{\lambda(|u-v|)}(a^p, b^p) \quad \text{where} \quad H_c(a^p, b^p) \overset{\text{def.}}{=} \inf_{\theta \geq 0} \theta L_c(\tfrac{a^p}{\theta}, \tfrac{b^p}{\theta})$$

is the perspective transform of $L_c$ introduced in Lemma 1. The intuition underpinning the definition of this cost is that the perspective transform accounts for the possibility to rescale a transport plan $\pi$ by a scalar $\theta$ but the scaling is performed pointwise instead of globally. In general $\mathcal{D}$ is not a distance, but it is always definite as stated by this result proved in Appendix A.

**Proposition 3.** *Assume $\lambda^{-1}(\{0\}) = \{0\}$, $\varphi^{-1}(\{0\}) = \{1\}$ and $\varphi$ is coercive. Then $\mathcal{D}$ is definite on $\mathfrak{C}[\mathbb{R}^+]$, i.e. $\mathcal{D}([u,a],[v,b]) = 0$ if and only if $(a = b = 0)$ or $(a = b$ and $u = v)$.*

Of particular interest are those $\varphi$ where $\mathcal{D}$ is a distance, which necessitates a careful choice of $\lambda, p$ and $q$. We now detail three examples where this is the case.

**Gaussian Hellinger distance (GH).** When $D_\varphi = \mathrm{KL}$, $\lambda(t) = t^2$ and $q = p = 2$, then one has $\mathcal{D}([u,a],[v,b])^2 = a^2 + b^2 - 2abe^{-|u-v|/2}$. This cone distance (Burago et al., 2001) is further generalized by De Ponti (2019) who shows that $\mathcal{D}$ is a distance for power entropies $\varphi(s) = \frac{s^p - p(s-1) - 1}{p(p-1)}$ if $p \geq 1$ (the case $p = 1$ corresponding to $D_\varphi = \mathrm{KL}$).

**Hellinger-Kantorovich (HK) / Wasserstein-Fisher-Rao distance (WFR).** When $D_\varphi = \mathrm{KL}$, $\lambda(t) = -\log \cos^2(t \wedge \tfrac{\pi}{2})$ and $q = p = 2$, then one has $\mathcal{D}([u,a],[v,b])^2 = a^2 + b^2 - 2ab \cos(\tfrac{\pi}{2} \wedge |u - v|)$. This construction, which might seem peculiar, corresponds to the one used to make unbalanced OT a geodesic distance, as detailed in (Liero et al., 2015; Chizat et al., 2018c).

**Partial optimal transport distance (PT).** When $D_\varphi = \mathrm{TV}$, $\lambda(t) = t^q$, $q \geq 1$ and $p = 1$, then $\mathcal{D}([u,a],[v,b])^q = a + b - (a \wedge b)(2 - |u - v|^q)_+$ defines a cone distance (Chizat et al., 2018c).

### 2.2.2 DEFINITIONS AND PROPERTIES

The conic formulation consists in solving a GW problem on the cone, with the addition of two linear constraints. Informally speaking, $L_c$ from Lemma 1 becomes $\mathcal{D}$, the term $(|(\mu \otimes \mu)^\perp| + |(\nu \otimes \nu)^\perp|)$ is taken into account by the constraints (5) below, and the variables $(f,g)$ are replaced by $(r^p, s^p)$. It reads $\mathrm{CGW}(\mathcal{X}, \mathcal{Y}) \overset{\text{def.}}{=} \inf_{\alpha \in \mathcal{U}_p(\mu,\nu)} \mathcal{H}(\alpha)$ where

$$\mathcal{H}(\alpha) \overset{\text{def.}}{=} \int \mathcal{D}([d_X(x,x'), rr'], [d_Y(y,y'), ss'])^q \mathrm{d}\alpha([x,r],[y,s]) \mathrm{d}\alpha([x',r'],[y',s']),$$

$$\mathcal{U}_p(\mu,\nu) \overset{\text{def.}}{=} \left\{ \alpha \in \mathcal{M}_+(\mathfrak{C}[X] \times \mathfrak{C}[Y]) \ : \ \int_{\mathbb{R}_+} r^p \mathrm{d}\alpha_1(\cdot, r) = \mu, \int_{\mathbb{R}_+} s^p \mathrm{d}\alpha_2(\cdot, s) = \nu \right\}. \tag{5}$$

It is similar to the conic formulation of UW, see Appendix A. Note that similarly to the GW formulation (2) – and in sharp contrast with the conic formulation of UW – here the transport plans are defined on the cone $\mathfrak{C}[X] \times \mathfrak{C}[Y]$ but the cost $\mathcal{D}$ is a distance on $\mathfrak{C}[\mathbb{R}_+]$.

We present now the main contributions of this paper, proved in Appendix C. We state that CGW defines a distance under conditions that hold for the settings of Section 2.2.1, and that it is upper-bounded by UGW. While the distance $\mathrm{CGW}^{1/q}$ cannot be casted as a finite dimensional program even in discrete settings (because it is defined on a lifted space), UGW can be approximated with efficient numerical schemes as detailed in Section 3. The tightness of the bound between UGW and CGW and the computation of CGW are open questions left for future works.

**Theorem 1.** *(i) The divergence CGW is symmetric, positive and definite up to isometries. (ii) If $\mathcal{D}$ is a distance on $\mathfrak{C}[\mathbb{R}_+]$, then $\mathrm{CGW}^{1/q}$ is a distance on the set of mm-spaces up to isometries. (iii) For any $(D_\varphi, \lambda, p, q)$ with associated cost $\mathcal{D}$ on the cone, one has $\mathrm{UGW} \geq \mathrm{CGW}$.*

**Sketch of proof** **(i)** CGW is positive and symmetric. Definiteness holds thanks to Proposition 3. **(ii)** The triangle inequality is similar to balanced OT, and applies the gluing lemma (Villani, 2003, Lemma 7.6). The non-trivial part is showing that the latter lemma holds, because it glues two plans provided they have a common marginal. Since CGW is invariant under radial rescalings (called dilations in Appendix C), it is possible to dilate two plans such that they have a common marginal

and remain optimal. **(iii)** Take an optimal plan $\pi$ for UGW$(\mathcal{X}, \mathcal{Y})$. From this $\pi$ one can build a plan $\alpha$ such that $\mathcal{L}(\pi) \geq \mathcal{H}(\alpha)$ because $L_c \geq H_c$. Furthermore $\alpha \in \mathcal{U}_p$, and is thus admissible and suboptimal, which yields UGW$(\mathcal{X}, \mathcal{Y}) = \mathcal{L}(\pi) \geq \mathcal{H}(\alpha) \geq$ CGW$(\mathcal{X}, \mathcal{Y})$.

## 3 ALGORITHMS

The computation of the distance CGW is in practice out-of-reach because it requires an optimization over a lifted conic space, which would need to be discretized. We focus in this section on the numerical computation of the upper bound UGW, using an alternate minimization coupled with entropic regularization. The algorithm is presented on arbitrary measures, the special case of discrete measures being a particular case. The discretized formulas and algorithms are detailed in Appendix D, see also Chizat et al. (2018a); Peyré et al. (2016). All implementations are available at https://github.com/anonymous-conference-submission.

In order to derive a simple numerical approximation scheme, following Mémoli (2011), we introduce a lower bound obtained by introducing two transportation plans. To further accelerate the method and enable GPU-friendly iterations, similarly to Gold et al. (1996); Solomon et al. (2016), we consider an entropic regularization. It reads, for any $\varepsilon \geq 0$,

$$\text{UGW}_\varepsilon(\mathcal{X}, \mathcal{Y}) \stackrel{\text{def.}}{=} \inf_\pi \mathcal{L}(\pi) + \varepsilon \text{KL}^\otimes(\pi | \mu \otimes \nu) \geq \inf_{\pi, \gamma} \mathcal{F}(\pi, \gamma) + \varepsilon \text{KL}(\pi \otimes \gamma | (\mu \otimes \nu)^{\otimes 2}), \tag{6}$$

$$\text{and} \quad \mathcal{F}(\pi, \gamma) \stackrel{\text{def.}}{=} \int_{X^2 \times Y^2} \lambda(|d_X - d_Y|) \mathrm{d}\pi \otimes \gamma + \mathrm{D}_\varphi(\pi_1 \otimes \gamma_1 | \mu \otimes \mu) + \mathrm{D}_\varphi(\pi_2 \otimes \gamma_2 | \nu \otimes \nu),$$

where $(\gamma_1, \gamma_2)$ denote the marginals of the plan $\gamma$. Note that in contrast to the entropic regularization of GW Peyré et al. (2016), here we use a tensorized entropy to maintain the overall homogeneity of the energy. A simple method to approximate this lower bound is to perform an alternate minimization on $\pi$ and $\gamma$, which is known to converge for smooth $\varphi$ to a stationary point since the coupling term in the functional is smooth (Tseng, 2001).

Note that if $\pi \otimes \gamma$ is optimal then so is $(s\pi) \otimes (\frac{1}{s}\gamma)$ with $s \geq 0$. Thus without loss of generality we optimize under the constraint $m(\pi) = m(\gamma)$ by setting $s = \sqrt{m(\gamma)/m(\pi)}$. In general, this bound is not expected to be tight, but empirically, alternate minimization often converges to a solution with $\pi = \gamma$ (as already observed for instance in Rangarajan et al. (1999); Solomon et al. (2016)), so that the algorithm also finds a local minimizer of the UGW$_\varepsilon$ problem. In the Balanced-GW case in Euclidean spaces, the optimum is known to satisfy $\pi = \gamma$ (Konno, 1976) and alternate descent is equivalent to a mirror descent algorithm (Solomon et al., 2016). Minimizing the lower bound of (6) with respect to either $\pi$ or $\gamma$ is non-trivial for an arbitrary $\varphi$. We restrict our attention to the Kullback-Leibler case $\mathrm{D}_\varphi = \rho\text{KL}$ with $\rho > 0$, which can be addressed by solving a regularized and convex unbalanced problem as studied in Chizat et al. (2018a); Séjourné et al. (2019). It is explained in the following proposition.

**Proposition 4.** *For a fixed $\gamma$, the optimal $\pi \in \arg\min_\pi \mathcal{F}(\pi, \gamma) + \varepsilon\text{KL}(\pi \otimes \gamma | (\mu \otimes \nu)^{\otimes 2})$ is the solution of $\min_\pi \int c_\gamma^\varepsilon(x, y)\mathrm{d}\pi(x, y) + \rho m(\gamma)\text{KL}(\pi_1|\mu) + \rho m(\gamma)\text{KL}(\pi_2|\nu) + \varepsilon m(\gamma)\text{KL}(\pi|\mu \otimes \nu)$, where $m(\gamma) \stackrel{\text{def.}}{=} \gamma(X \times Y)$ is the mass of $\gamma$, and where we define the cost associated to $\gamma$ as $c_\gamma^\varepsilon(x, y) \stackrel{\text{def.}}{=} \int \lambda(|d_X(x, \cdot) - d_Y(y, \cdot)|)\mathrm{d}\gamma + \rho \int \log(\frac{\mathrm{d}\gamma_1}{\mathrm{d}\mu})\mathrm{d}\gamma_1 + \rho \int \log(\frac{\mathrm{d}\gamma_2}{\mathrm{d}\nu})\mathrm{d}\gamma_2 + \varepsilon \int \log(\frac{\mathrm{d}\gamma}{\mathrm{d}\mu\mathrm{d}\nu})\mathrm{d}\gamma$.*

Computing the cost $c_\gamma^\varepsilon$ for spaces $X$ and $Y$ of $n$ points has in general a cost $O(n^4)$ in time and memory. However, as explained for instance in Peyré et al. (2016), for the special case $\lambda(t) = t^2$, this cost is reduced to $O(n^3)$ in time and $O(n^2)$ in memory. This is the setting we consider in the numerical simulations. This makes the method applicable for scales of the order of $10^4$ points. For larger datasets one should use approximation schemes such as hierarchical approaches (Xu et al., 2019) or Nyström compression of the kernel (Altschuler et al., 2018).

The resulting alternate minimization method is detailed in Algorithm 1, see Appendix D for a discretized version. It uses the unbalanced Sinkhorn algorithm of Chizat et al. (2018a); Séjourné et al. (2019) as sub-iterations and is initialized using $\pi = \mu \otimes \nu / \sqrt{m(\mu)m(\nu)}$. This Sinkhorn algorithm operates over a pair of continuous functions (so-called Kantorovich potentials) $f(x)$ and $g(y)$. For discrete spaces $X$ and $Y$ of size $n$, these functions are stored in vectors of size $n$, and that integral

---

**Algorithm 1 – UGW($\mathcal{X}, \mathcal{Y}, \rho, \varepsilon$)**

---

**Input:** mm-spaces $(\mathcal{X}, \mathcal{Y})$, relaxation $\rho$, regularization $\varepsilon$
**Output:** approximation $(\pi, \gamma)$ minimizing 6

1: Initialize $\pi = \gamma = \mu \otimes \nu / \sqrt{m(\mu)m(\nu)}$, $g = 0$.
2: **while** $(\pi, \gamma)$ has not converged **do**
3:     Update $\pi \leftarrow \gamma$, then $c \leftarrow c_\pi^\varepsilon$, $\tilde{\rho} \leftarrow m(\pi)\rho$, $\tilde{\varepsilon} \leftarrow m(\pi)\varepsilon$
4:     **while** $(f, g)$ has not converged **do**
5:         $\forall x, \; f(x) \leftarrow -\frac{\tilde{\varepsilon}\tilde{\rho}}{\tilde{\varepsilon}+\tilde{\rho}} \log\left( \int e^{(g(y)-c(x,y))/\tilde{\varepsilon}} \mathrm{d}\nu(y) \right)$
6:         $\forall y, \; g(y) \leftarrow -\frac{\tilde{\varepsilon}\tilde{\rho}}{\tilde{\varepsilon}+\tilde{\rho}} \log\left( \int e^{(f(x)-c(x,y))/\tilde{\varepsilon}} \mathrm{d}\mu(x) \right)$
7:     Update $\gamma(x,y) \leftarrow \exp\left[ (f(x)+g(y)-c(x,y))/\tilde{\varepsilon} \right] \mu(x)\nu(y)$
8:     Rescale $\gamma \leftarrow \sqrt{m(\pi)/m(\gamma)}\gamma$
9: Return $(\pi, \gamma)$.

---

involved in the updates becomes a sum. Each iteration of Sinkhorn thus has a cost $n^2$, and all the involved operation can be efficiently mapped to parallelizable GPU routines as detailed in Chizat et al. (2018a); Séjourné et al. (2019). Another advantage of using an unbalanced Sinkhorn algorithm is its complexity $O(n^2/\varepsilon)$ to compute an $\varepsilon$-approximation, as stated in Pham et al. (2020), which should be compared to $O(n^2/\varepsilon^2)$ operations for balanced Sinkhorn.

Note also that balanced GW is recovered as a special case when setting $\rho \to +\infty$, so that $\tilde{\rho}/(\tilde{\varepsilon} + \tilde{\rho}) \to 1$ should be used in the iterations. In order to speed up Sinkhorn inner-loops, especially for small values of $\varepsilon$, one can use linear extrapolation (Thibault et al., 2017) or non-linear Anderson acceleration (Scieur et al., 2016).

There is an extra scaling step after computing $\gamma$ involving the mass $m(\pi)$. It corresponds to the scaling $s$ of $\pi \otimes \gamma$ such that $m(\pi) = m(\gamma)$, and we observe that this scaling is key not only to impose this mass equality but also to stabilize the algorithm. Otherwise we observed that $m(\gamma) < 1 < m(\pi)$ and underflows whenever $m(\gamma) \to 0$ and $m(\pi) \to \infty$.

## 4 NUMERICAL EXPERIMENTS

This section presents numerical simulations on synthetic examples, to highlight the qualitative behavior of UGW with respect to mass variation and outliers. In all these experiments, $\mu$ and $\nu$ are probability distributions, which allows us to compare GW with UGW.

**Robustness to imbalanced classes.** In this first example, we take $X = Y = \mathbb{R}^2$ and consider $\mathcal{E}, \mathcal{C}$ and $\mathcal{S}$ to be uniform distributions on an ellipse, a disk and a square. Figure 1 contrasts the transportation plan obtained by GW and UGW for a fixed $\mu = 0.5\mathcal{E} + 0.5\mathcal{C}$ and $\nu$ obtained using two different mixtures of $\mathcal{E}$ and $\mathcal{S}$. The black segments show the largest entries of the transportation matrix $\pi$, for a sub-sampled set of points (to ease visibility), thus effectively displaying the matching induced by the plan. Furthermore, the width of the dots are scaled according to the mass of the marginals $\pi_1 \approx \mu$ and $\pi_2 \approx \nu$, i.e. the smaller the point, the smaller is the amount of transported mass. This figure shows that the exact conservation of mass imposed by GW leads to a poor geometrical matching of the shapes which have different global mass. As this should be expected, UGW recovers coherent matchings. We suspect the alternate minimization algorithm was able to find the global minimum in these cases.

**Influence of $\varepsilon$ and debiasing.** This figure (and the following ones) does not show the influence of $\varepsilon$. This parameter is set of a low value $\varepsilon = 10^{-2}$ on a domain $[0,1]^2$ so as to approximate the optimal plan of the unregularized UGW problem . The impact of $\varepsilon$ is similar to those of classical OT, namely that it introduces an extra diffusion bias.

**Robustness to outlier** Figure 2 shows another experiment on a 2-D dataset, using the same display convention as Figure 1. It corresponds to the two moons dataset with additional outliers (displayed

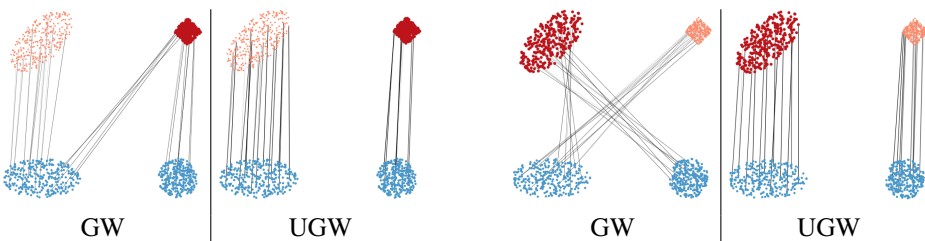

Figure 1: GW vs. UGW transportation plan, using $\nu = 0.3\mathcal{E} + 0.7\mathcal{S}$ on the left, and $\nu = 0.7\mathcal{E} + 0.3\mathcal{S}$ on the right.

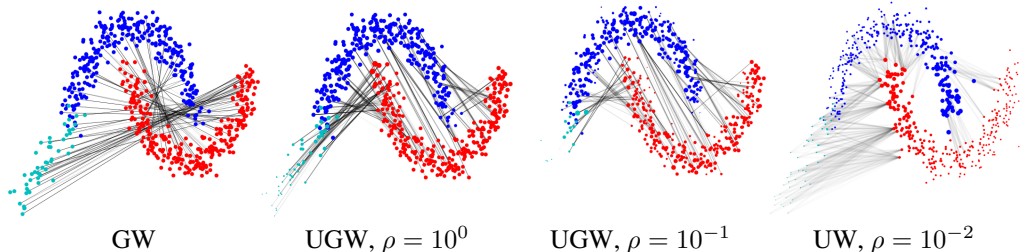

Figure 2: GW and UGW applied to two moons with outliers.

in cyan). Decreasing the value of $\rho$ (thus allowing for more mass creation/destruction in place of transportation) is able to reduce and even remove the influence of the outliers, as expected. Furthermore, using small values of $\rho$ tends to favor "local structures", which is a behavior quite different from UW (1). Indeed, for UW, $\rho \to 0$ sets to zero all the mass of $\pi$ outside of the diagonal (points are not transported), while for UGW, it is rather pairs of points with dissimilar pairwise distances which cannot be transported together.

**Graph matching and comparison with Partial-GW.** We now consider two graphs $(X, Y)$ equipped with their respective geodesic distances. These graphs correspond to points embedded in $\mathbb{R}^2$, and the length of the edges corresponds to their Euclidean length. These two synthetic graphs are close to be isometric, but differ by addition or modification of small sub-structures. The colors $c(x)$ are defined on the "source" graph $X$ and are mapped by an optimal plan $\pi$ on $y \in Y$ to a color $\frac{1}{\pi_1(y)} \int_X c(x) \mathrm{d}\pi(x, y)$. This allows to visualize the matching induced by GW and UGW for a varying $\rho$, as displayed in Figure 3. The graphs for GW should be taken as reference since there is no mass creation. The POT library (Flamary & Courty, 2017) is used to compute GW.

For large values of $\rho$, UGW behaves similarly to GW, thus producing irregular matchings which do not preserve the overall geometry of the shapes. In sharp contrast, for smaller values of $\rho$ (e.g. $\rho = 10^{-1}$), some fine scale structures (such as the target's small circle) are discarded, and UGW is able to produce a meaningful partial matching of the graphs. For intermediate values ($\rho = 10^0$), we observe that the two branches and the blue cluster of the source are correctly matched to the target, while for GW the blue points are scattered because of the marginal constraint.

Figure 4 shows a comparison with Partial-GW (Chapel et al., 2020), computed using the POT library. It is close to UGW with a $\mathrm{TV}^\otimes$ penalty, since partial OT is equivalent to the use of a TV relaxation of the marginal. UGW with a $\mathrm{KL}^\otimes$ penalty is first computed for a given $\rho$, then the total mass $m$ of the optimal plan is computed, and is used as a parameter for PGW which imposes this total mass as a constraint. Figure 3 and 4 display the transportation strategy associated to both methods. KL-UGW operates smooth transitions between transportation and creation of mass, while PGW either performs pure transportation or pure destruction/creation of mass. This can be observed in Figure 4 where nodes of the graphs are removed and not taken into account by the matching. Note also that since PGW is equivalent to solving GW on sub-graphs, the color distribution of GW and PGW are the same.

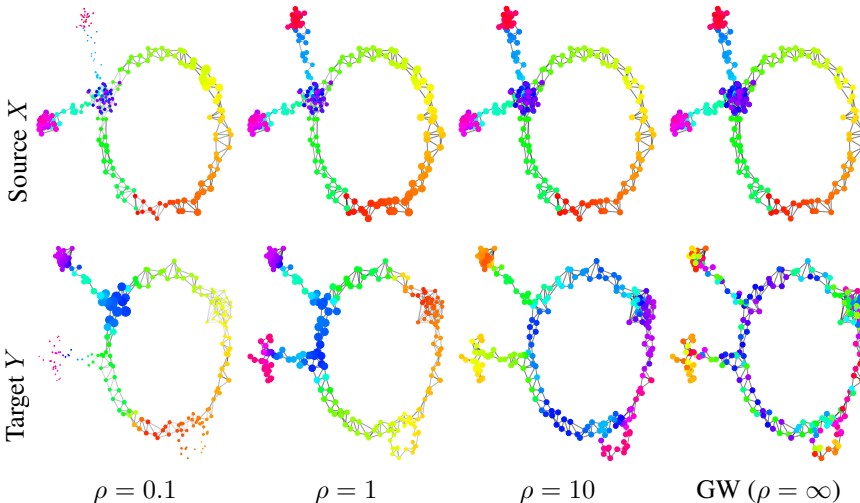

Figure 3: Comparison of UGW and GW for graph matching.

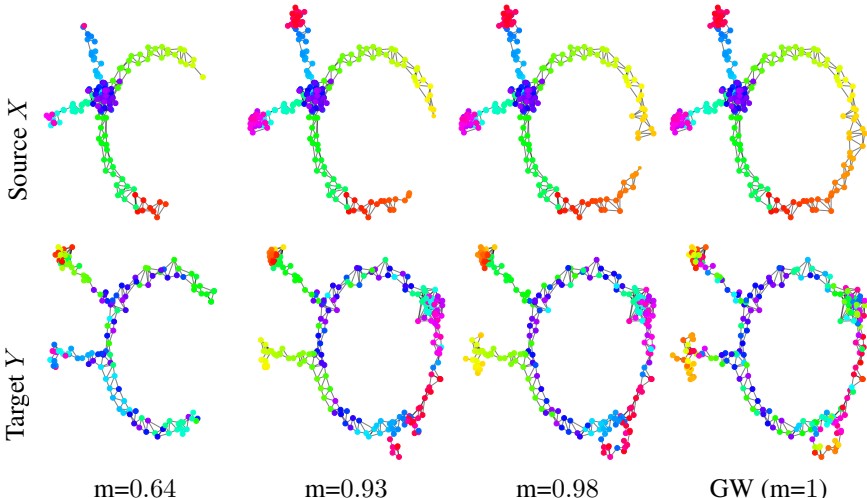

Figure 4: Comparison of Partial-GW for graph matching. Here $m$ is the budget of transported mass.

## 5 CONCLUSION AND PERSPECTIVES

This paper defines two Unbalanced Gromov-Wasserstein formulations. We prove that they are both positive and definite. We provide a scalable, GPU-friendly algorithm to compute one of them, and show that the other is a distance between mm-spaces up to isometry. These divergences and distances allow for the first time to blend in a seamless way the transportation geometry of GW with creation and destruction of mass. This hybridization is the key to unlock both theoretical and practical issues. This work opens new questions for futures works. On the theoretical side, the geodesic structures induced by unbalanced GW distances and divergences is an important subject of study. On the practical side, removing the bias introduced by the use of entropic regularization is important for applications to ML. Note that such a debiasing was successfully applied for Balanced-GW in Bunne et al. (2019) and is shown to lead to a valid divergence for balanced OT in Feydy et al. (2019b) and UW in Séjourné et al. (2019). The design of efficient numerical solvers for the the conic formulation is also an interesting avenue for future works.

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

# A  BACKGROUND ON UNBALANCED OPTIMAL TRANSPORT

Following Liero et al. (2015), this section reviews and generalizes the homogeneous and conic formulations of unbalanced optimal transport. These three formulations are equal in the convex setting of UOT. Our relaxed divergence UGW and conic distance CGW defined in Section 2 build upon those constructions but are not anymore equal due to the non-convexity of GW problems.

## A.1  HOMOGENEOUS FORMULATION

To ease the description of the homogeneous formulation, we develop and refactor the Csiszàr divergence terms of (1) in a form analog to Lemma 1. It reads

$$\mathrm{UW}(\mu, \nu)^q = \inf_{\pi \in \mathcal{M}(X^2)} \int L_{\lambda(d(x,y))}(f(x), g(y))\mathrm{d}\pi(x, y) + \psi'_\infty(|\mu^\perp| + |\nu^\perp|), \qquad (7)$$

where $L_c(r, s) \stackrel{\mathrm{def.}}{=} c + r\varphi(1/r) + s\varphi(1/s)$, $|\mu^\perp| \stackrel{\mathrm{def.}}{=} \mu^\perp(X)$ and $(f \stackrel{\mathrm{def.}}{=} \frac{\mathrm{d}\mu}{\mathrm{d}\pi_1}, g \stackrel{\mathrm{def.}}{=} \frac{\mathrm{d}\nu}{\mathrm{d}\pi_2})$ are the densities of the Lebesgue decomposition of $(\mu, \nu)$ with respect to $(\pi_1, \pi_2)$ and

$$\mu = f\pi_1 + \mu^\perp \quad \text{and} \quad \nu = g\pi_2 + \nu^\perp. \qquad (8)$$

Such form is helpful to explicit the terms of pure mass creation/destruction $(|\mu^\perp| + |\nu^\perp|)$ and reinterpret the integral under $\pi$ as a transport term with a new cost $L_{\lambda(d)}$.

Then the authors of Liero et al. (2015) define the homogeneous formulations HUW as

$$\mathrm{HUW}(\mu, \nu)^q \stackrel{\mathrm{def.}}{=} \inf_{\pi \in \mathcal{M}(X^2)} \int H_{\lambda(d(x,y))}(f(x), g(y))\mathrm{d}\pi(x, y) + \psi'_\infty(|\mu^\perp| + |\nu^\perp|), \qquad (9)$$

where the 1-homogeneous function $H_c$ is the perspective transform of $L_c$

$$H_c(r, s) \stackrel{\mathrm{def.}}{=} \inf_{\theta \geq 0} \theta\big(c + \psi(\tfrac{r}{\theta}) + \psi(\tfrac{s}{\theta})\big) = \inf_{\theta \geq 0} \theta L_c(\tfrac{r}{\theta}, \tfrac{s}{\theta}). \qquad (10)$$

By definition one has $L_c \geq H_c$, though after optimization one has UW = HUW.

## A.2  CONE SETS, CONE DISTANCES AND EXPLICIT SETTINGS

The conic formulation detailed in Section A.3 is obtained by performing the optimal transport on the cone set $\mathfrak{C}[X] \stackrel{\mathrm{def.}}{=} X \times \mathbb{R}_+/(X \times \{0\})$, where the extra coordinate accounts for the mass of the particle. Coordinates of the form $(x, 0)$ are merged into a single point called the apex of the cone, noted $\mathfrak{o}_X$. In the sequel, points of $X \times \mathbb{R}_+$ are noted $(x, r)$ and those of $\mathfrak{C}[X]$ are noted $[x, r]$ to emphasize the quotient operation at the apex.

For a pair $(p, q) \in \mathbb{R}_+$, we define for any $[x, r], [y, s] \in \mathfrak{C}[X]^2$

$$\mathcal{D}_{\mathfrak{C}[X]}([x, r], [y, s])^q \stackrel{\mathrm{def.}}{=} H_{\lambda(d(x,y))}(r^p, s^p). \qquad (11)$$

In general $\mathcal{D}_{\mathfrak{C}[X]}$ is not a distance, but it is always definite as proved by the following result.

**Proposition 5.** *Assume that $d$ is definite, $\lambda^{-1}(\{0\}) = \{0\}$ and $\varphi^{-1}(\{0\}) = \{1\}$. Assume also that for any $(r, s)$, there always exists $\theta^*$ such that $H_c(r, s) = \theta^* L_c(\frac{r}{\theta^*}, \frac{s}{\theta^*})$. Then $\mathcal{D}_{\mathfrak{C}[X]}$ is definite on $\mathfrak{C}[X]$, i.e. $\mathcal{D}_{\mathfrak{C}[X]}([x, r], [y, s]) = 0$ if and only if $(r = s = 0)$ or $(r = s$ and $x = y)$.*

*Proof.* Assume $\mathcal{D}_{\mathfrak{C}[X]}([x, r], [y, s]) = 0$, and write $\theta^*$ such that

$$\mathcal{D}_{\mathfrak{C}[X]}([x, r], [y, s])^q = \theta^* L_c(\tfrac{r^p}{\theta^*}, \tfrac{s^p}{\theta^*}) = \theta^* \lambda(d(x, y)) + r^p \varphi(\tfrac{\theta^*}{r^p}) + s\varphi(\tfrac{\theta^*}{s^p}),$$

where the last line is given by the definition of reverse entropy. There are two cases. If $\theta^* > 0$, since all terms are positive, there are all equal to 0. By definiteness of $d$ it yields $x = y$ and because $\varphi^{-1}(\{0\}) = \{1\}$ we have $r^p = s^p = \theta^*$ and $r = s$. If $\theta^* = 0$ then $\mathcal{D}_{\mathfrak{C}[X]}([x, r], [y, s])^q = \varphi(0)(r^p + s^p)$. The assumption $\varphi^{-1}(\{0\}) = \{1\}$ implies $\varphi(0) > 0$, thus necessarily $r = s = 0$.  □

The function $H_c$ can be computed in closed form for a certain number of common entropies $\varphi$, and we refer to Liero et al. (2015, Section 5) for an overview. Of particular interest are those $\varphi$ where $\mathcal{D}_{\mathfrak{C}[X]}$ is a distance, which necessitates a careful choice of $\lambda, p$ and $q$. We now detail three particular settings where this is the case. In each setting we provide $(\mathrm{D}_\varphi, \lambda, p, q)$ and its associated cone distance $\mathcal{D}_{\mathfrak{C}[X]}$.

**Gaussian Hellinger distance**    It corresponds to

$$\mathrm{D}_\varphi = \mathrm{KL}, \quad \lambda(t) = t^2 \quad \text{and} \quad q = p = 2,$$
$$\mathcal{D}_{\mathfrak{C}[X]}([x,r],[y,s])^2 = r^2 + s^2 - 2rse^{-d(x,y)/2},$$

in which case it is proved in Liero et al. (2015) that $\mathcal{D}_{\mathfrak{C}[X]}$ is a cone distance.

**Hellinger-Kantorovich / Wasserstein-Fisher-Rao distance**    It reads

$$\mathrm{D}_\varphi = \mathrm{KL}, \quad \lambda(t) = -\log\cos^2(t \wedge \tfrac{\pi}{2}) \quad \text{and} \quad q = p = 2,$$
$$\mathcal{D}_{\mathfrak{C}[X]}([x,r],[y,s])^2 = r^2 + s^2 - 2rs\cos(\tfrac{\pi}{2} \wedge d(x,y)),$$

in which case it is proved in Burago et al. (2001) that $\mathcal{D}_{\mathfrak{C}[X]}$ is a cone distance.

The weight $\lambda(t) = -\log\cos^2(t \wedge \frac{\pi}{2})$, which might seem more peculiar, is in fact the penalty that makes unbalanced OT a length space induced by the Gaussian-Hellinger distance (if the ground metric $d$ is itself geodesic), as proved in Liero et al. (2016); Chizat et al. (2018b). This weight introduces a cut-off, because $\lambda(d(x,y)) = +\infty$ if $d(x,y) > \pi/2$. There is no transport between points too far from each other. The choice of $\pi/2$ is arbitrary, and can be modified by scaling $\lambda \mapsto \lambda(\cdot/s)$ for some cutoff $s$.

**Partial optimal transport**    It corresponds to

$$\mathrm{D}_\varphi = \mathrm{TV}, \quad \lambda(t) = t^q \quad \text{and} \quad q \geq 1 \quad \text{and} \quad p = 1,$$
$$\mathcal{D}_{\mathfrak{C}[X]}([x,r],[y,s])^q = r + s - (r \wedge s)(2 - d(x,y)^q)_+,$$

in which case it is proved in Chizat et al. (2018c) that $\mathcal{D}_{\mathfrak{C}[X]}$ is a cone distance. The case $\mathrm{D}_\varphi = \mathrm{TV}$ is equivalent to partial unbalanced OT, which produces discontinuities (because of the non-smoothness of the divergence) between regions of the supports which are being transported and regions where mass is being destroyed/created. Note that Liero et al. (2015) do not mention that this $\mathcal{D}_{\mathfrak{C}[X]}$ defines a distance, so this result is new to the best of our knowledge, although it can be proved without a conic lifting that partial OT defines a distance as explained in Chizat et al. (2018c).

A.3    CONIC FORMULATION OF UW

The last formulation reinterprets UW as an OT problem on the cone, with the addition of two linear constraints. Informally speaking, $H_c$ becomes $\mathcal{D}_{\mathfrak{C}[X]}$, the term $(|\mu^\perp| + |\nu^\perp|)$ is taken into account by the constraints (13) below, and the variables $(f,g)$ are replaced by $(r^p, s^p)$. It reads

$$\mathrm{CUW}(\mu,\nu)^q \overset{\text{def.}}{=} \inf_{\alpha \in \mathcal{U}_p(\mu,\nu)} \int \mathcal{D}_{\mathfrak{C}[X]}([x,r],[y,s]))^q \mathrm{d}\alpha([x,r],[y,s]), \tag{12}$$

where the constraint set $\mathcal{U}_p(\mu,\nu)$ is defined as

$$\mathcal{U}_p(\mu,\nu) \overset{\text{def.}}{=} \left\{ \alpha \in \mathcal{M}_+(\mathfrak{C}[X]^2) \ : \ \int_{\mathbb{R}_+} r^p \mathrm{d}\alpha_1(\cdot,r) = \mu, \int_{\mathbb{R}_+} s^p \mathrm{d}\alpha_2(\cdot,s) = \nu \right\}. \tag{13}$$

Thus CUW consists in minimizing the Wasserstein distance $\mathrm{W}_{\mathcal{D}_{\mathfrak{C}[X]}}(\alpha_1,\alpha_2)$ on the cone $(\mathfrak{C}[X], \mathcal{D}_{\mathfrak{C}[X]})$. The additional constraints on $(\alpha_1,\alpha_2)$ mean that the lift of the mass on the cone must be consistent with the total mass of $(\mu,\nu)$. When $\mathcal{D}_{\mathfrak{C}[X]}$ is a distance, CUW inherits the metric properties of $\mathrm{W}_{\mathcal{D}_{\mathfrak{C}[X]}}$. Our theoretical results rely on an analog construction for GW.

The following proposition states the equality of the three formulations and recapitulates its main properties. The proofs are detailed in Liero et al. (2015).

**Proposition 6** (From Liero et al. (2015)). *One has* UW = HUW = CUW, *which are symmetric, positive and definite. Furthermore, if* $(X, d_X)$ *and* $(\mathfrak{C}[X], \mathcal{D}_{\mathfrak{C}[X]})$ *are metric spaces with* $X$ *separable, then* $\mathcal{M}_+(X)$ *endowed with* CUW *is a metric space.*

*Proof.* The equality UW = HUW is given by Liero et al. (2015, Theorem 5.8), while the equality HUW = CUW holds thanks to Liero et al. (2015, Theorem 6.7 and Remark 7.5), where the latter

theorem can be straightforwardly generalized to any cone distance built as in Section 2.2.1. Since $\mathcal{D}_{\mathfrak{C}[X]}$ is symmetric, positive and definite (see Proposition 3), then so is CUW. Furthermore, if $\mathcal{D}_{\mathfrak{C}[X]}$ satisfies the triangle inequality, separability of $X$ allows to apply the gluing lemma (Liero et al., 2015, Corollary 7.14) which generalizes to any exponent $p$ defining $\mathcal{U}_p(\mu, \nu)$ and any cone distance $\mathcal{D}_{\mathfrak{C}[X]}$. $\qquad\square$

## B UGW FORMULATION AND DEFINITENESS

We present in this section the proofs of the properties of our divergence UGW. We refer to Section 2 for the definition of the UGW formulation and its related concepts. For conciseness we write $\Gamma(x, x', y, y') = |d_X(x, x') - d_Y(y, y')|$.

We first start with the existence of minimizers stated in Proposition 1. It illustrates in some sense that our divergence is well-defined.

**Proposition 7** (Existence of minimizers). *Assume $(\mathcal{X}, \mathcal{Y})$ to be compact mm-spaces and that we either have*

1. *$\varphi$ superlinear, i.e $\varphi'_\infty = \infty$*

2. *$\lambda$ has compact sublevel sets in $\mathbb{R}_+$ and $2\varphi'_\infty + \inf \lambda > 0$*

*Then there exists $\pi \in \mathcal{M}_+(X \times Y)$ such that $\mathrm{UGW}(\mathcal{X}, \mathcal{Y}) = \mathcal{L}(\pi)$.*

*Proof.* We adapt here from Liero et al. (2015, Theorem 3.3). The functional is lower semi-continuous as a sum of l.s.c terms. Thus it suffices to have relative compactness of the set of minimizers. Under either one of the assumptions, coercivity of the functional holds thanks to Jensen's inequality

$$\mathcal{L}(\pi) \geq m(\pi)^2 \inf \lambda(\Gamma) + m(\mu)^2 \varphi\left(\frac{m(\pi)^2}{m(\mu)^2}\right) + m(\nu)^2 \varphi\left(\frac{m(\pi)^2}{m(\nu)^2}\right)$$

$$\geq m(\pi)^2 \left[ \inf \lambda(\Gamma) + \frac{m(\mu)^2}{m(\pi)^2} \varphi\left(\frac{m(\pi)^2}{m(\mu)^2}\right) + \frac{m(\nu)^2}{m(\pi)^2} \varphi\left(\frac{m(\pi)^2}{m(\nu)^2}\right) \right].$$

As $m(\pi) \to +\infty$ the right hand side converges to $2\varphi'_\infty + \inf \lambda > 0$, which under either one of the assumptions yields $\mathcal{L}(\pi) \to +\infty$, hence the coercivity. Thus we can assume there exists some $M$ such that $m(\pi) < M$. Since the spaces are assumed to be compact, the Banach-Alaoglu theorem holds and gives relative compactness in $\mathcal{M}_+(X \times Y)$.

Take any sequence of plans $\pi_n$ that approaches $\mathrm{UGW}(\mathcal{X}, \mathcal{Y}) = \inf \mathcal{L}(\pi)$. Compactness gives that a subsequence $\pi_{n_k}$ weak* converges to some $\pi^*$. Because $\mathcal{L}$ is l.s.c, we have $\mathcal{L}(\pi^*) \leq \inf \mathcal{L}(\pi)$, thus $\mathcal{L}(\pi^*) = \inf \mathcal{L}(\pi)$. The existence of such limit reaching the infimum gives the existence of a minimizer. $\qquad\square$

Note that this formulation is nonegative and symmetric because the functional $\mathcal{L}$ is also nonegative and symmetric in its inputs $(\mathcal{X}, \mathcal{Y})$. This formulation allows straightforwardly to prove the definiteness of UGW.

**Proposition 8** (Definiteness of UGW). *Assume that $\varphi^{-1}(\{0\}) = \{1\}$ and $\lambda^{-1}(\{0\}) = \{0\}$. The following assertions are equivalent:*

1. *$\mathrm{UGW}(\mathcal{X}, \mathcal{Y}) = 0$*

2. *$\exists \pi \in \mathcal{M}_+(X \times Y)$ whose marginals are $(\mu, \nu)$ such that $d_X(x, x') = d_Y(y, y')$ for $\pi \otimes \pi$-a.e. $(x, x', y, y') \in (X \times Y)^2$.*

3. *There exists a mm-space $(Z, d_Z, \eta)$ with full support and Borel maps $\psi_X : Z \to X$ and $\psi_Y : Z \to Y$. such that $(\psi_X)_\sharp \eta = \mu$, $(\psi_Y)_\sharp \eta = \nu$ and $d_Z = (\psi_X)^\sharp d_X = (\psi_Y)^\sharp d_Y$*

4. *There exists a Borel measurable bijection between the measures' supports $\psi : spt(\mu) \to spt(\nu)$ with Borel measurable inverse such that $\psi_\sharp \mu = \nu$ and $d_Y = \psi^\sharp d_X$.*

*Proof.* Recall that $(2) \Leftrightarrow (3) \Leftrightarrow (4)$ from Sturm (2012, Lemma 1.10). thus it remains to prove $(1) \Leftrightarrow (2)$.

If there is such coupling plan $\pi$ between $(\mu, \nu)$ then one has $\pi \otimes \pi$-a.e. that $\Gamma = 0$, and all $\varphi$-divergences are zero as well, yielding a distance of zero a.e.

Assume now that $\mathrm{UGW}(\mathcal{X}, \mathcal{Y}) = 0$, and write $\pi$ an optimal plan. All terms of $\mathcal{L}$ are positive, thus under our assumptions we have $\Gamma = 0$, $\pi_1 \otimes \pi_1 = \mu \otimes \mu$ and $\pi_2 \otimes \pi_2 = \nu \otimes \nu$. Thus we get that $\pi$ has marginals $(\mu, \nu)$ and that $d_X(x, x') = d_Y(y, y')$ $\pi \otimes \pi$-a.e. $\qquad\square$

We end with a result on the reformulation of UGW which is the first step to connnect it with the conic formulation CGW.

**Lemma 2.** *Defining* $L_c(r, s) \stackrel{\text{def.}}{=} c + r\varphi(1/r) + s\varphi(1/s)$, *and writing* $(f \stackrel{\text{def.}}{=} \frac{\mathrm{d}\mu}{\mathrm{d}\pi_1}, g \stackrel{\text{def.}}{=} \frac{\mathrm{d}\nu}{\mathrm{d}\pi_2})$ *the Lebesgue densities of* $(\mu, \nu)$ *w.r.t.* $(\pi_1, \pi_2)$ *such that* $\mu = f\pi_1 + \mu^\perp$ *and* $\nu = g\pi_2 + \nu^\perp$, *one has*

$$\mathcal{L}(\pi) = \int_{X^2 \times Y^2} L_{\lambda(\Gamma)}(f \otimes f, g \otimes g) \mathrm{d}\pi \mathrm{d}\pi + \varphi(0)(|(\mu \otimes \mu)^\perp| + |(\nu \otimes \nu)^\perp|).$$

*Proof.* Using Equation (23), one has

$$\begin{aligned}
\mathcal{L}(\pi) &= \int_{X^2 \times Y^2} \lambda(\Gamma) \mathrm{d}\pi \mathrm{d}\pi + \mathrm{D}_\varphi^\otimes(\pi_1 | \mu) + \mathrm{D}_\varphi^\otimes(\pi_2 | \nu) \\
&= \int_{X^2 \times Y^2} \lambda(\Gamma) \mathrm{d}\pi \mathrm{d}\pi + \mathrm{D}_\psi^\otimes(\mu | \pi_1) + \mathrm{D}_\psi^\otimes(\nu | \pi_2) \\
&= \int_{X^2 \times Y^2} \lambda(\Gamma) \mathrm{d}\pi \mathrm{d}\pi + \int_{X^2} \psi(f \otimes f) \mathrm{d}\pi_1 \mathrm{d}\pi_1 + \int_{Y^2} \psi(g \otimes g) \mathrm{d}\pi_2 \mathrm{d}\pi_2 \\
&\quad + \varphi(0)(|(\mu \otimes \mu)^\perp| + |(\nu \otimes \nu)^\perp|) \\
&= \int_{X^2 \times Y^2} L_{\lambda(\Gamma)}(f \otimes f, g \otimes g) \mathrm{d}\pi \mathrm{d}\pi + \varphi(0)(|(\mu \otimes \mu)^\perp| + |(\nu \otimes \nu)^\perp|).
\end{aligned}$$

$\qquad\square$

## C  CONIC FORMULATION AND METRIC PROPERTIES

We present in this section the proofs of the properties mentioned in Section 2. We refer to Section 2 and Appendix A for the definition of the conic formulation and its related concepts.

In this section we frequently use the notion of marginal for neasures. For any sets $E, F$, we write $\mathfrak{P}^{(E)} : E \times F \to E$ the **canonical projection** such that for any $(x, y) \in E \times F$, $\mathfrak{P}^{(E)}(x, y) = x$. Consider two complete separable mm-spaces $\mathcal{X} = (X, d_X, \mu)$ and $\mathcal{Y} = (Y, d_Y, \nu)$. Write $\pi \in \mathcal{M}_+(X \times Y)$ a coupling plan, and define its marginals by $\pi_1 = \mathfrak{P}_\sharp^{(X)} \pi$ and $\pi_2 = \mathfrak{P}_\sharp^{(Y)} \pi$. The definition of the marginals can also be seen by the use of test functions. In the case of $\pi_1$ it reads for any test function $\xi$

$$\int \xi(x) \mathrm{d}\pi_1(x) = \int \xi(x) \mathrm{d}\pi(x, y).$$

### C.1  PRELIMINARY RESULTS

We present in this section concepts and properties which are necessary for the proof of Theorem 1. We introduce a dilation operator whose role is to rescale the radial coordinate of a measure with a given scaling.

**Definition 2** (dilations)**.** *Consider* $v([x, r], [y, s])$ *a Borel measurable scaling function depending on* $[x, r], [y, s] \in \mathfrak{C}[X] \times \mathfrak{C}[Y]$. *Take a plan* $\alpha \in \mathcal{M}_+(\mathfrak{C}[X] \times \mathfrak{C}[Y])$. *We define the dilation* $\mathrm{Dil}_v : \alpha \mapsto (h_v)_\sharp(v^p \alpha)$ *where*

$$h_v([x, r], [y, s]) \stackrel{\text{def.}}{=} ([x, r/w], [y, s/w]),$$

*where $w = v([x, r], [y, s])$. It reads for any test function $\xi$*

$$\int \xi([x, r], [y, s]) \mathrm{dDil}_v(\alpha) = \int \xi([x, r/w], [y, s/w]) w^p \mathrm{d}\alpha.$$

The importance of dilations is given by the following lemma.

**Lemma 3** (Invariance to dilation). *The problem* CGW *is invariant to dilations, i.e. for any $\alpha \in \mathcal{U}_p(\mu, \nu)$, we have $\mathrm{Dil}_v(\alpha) \in \mathcal{U}_p(\mu, \nu)$ and $\mathcal{H}(\alpha) = \mathcal{H}(\mathrm{Dil}_v(\alpha))$.*

*Proof.* First we prove the stability of $\mathcal{U}_p(\mu, \nu)$ under dilations. Take $\alpha \in \mathcal{U}_p(\mu, \nu)$. For any test function $\xi$ defined on $X$ we have

$$\int \xi(x) r^p \mathrm{dDil}_v(\alpha) = \int \xi(x) (\frac{r}{v})^p . v^p \mathrm{d}(\alpha) = \int \xi(x) r^p \mathrm{d}\alpha = \int \xi(x) \mathrm{d}\mu(x).$$

Similarly we get $\mathfrak{P}_\sharp^{(Y)}(s^q \mathrm{Dil}_v(\alpha)) = \nu$, thus $\mathrm{Dil}_v(\alpha) \in \mathcal{U}_p(\mu, \nu)$.

It remains to prove the invariance of the functional. Recall that $\mathcal{D}^q$ is p-homogeneous. It yields

$$\mathcal{H}(\mathrm{Dil}_v(\alpha)) = \int \mathcal{D}([d_X(x, x'), rr'], [d_Y(y, y'), ss'])^q \mathrm{dDil}_v(\alpha) \mathrm{dDil}_v(\alpha)$$

$$= \int \mathcal{D}([d_X(x, x'), \frac{r}{v} \cdot \frac{r'}{v}], [d_Y(y, y'), \frac{s}{v} \cdot \frac{s'}{v}])^q v^p \cdot v^p \mathrm{d}\alpha \mathrm{d}\alpha$$

$$= \int \frac{1}{v^{2p}} \mathcal{D}([d_X(x, x'), rr'], [d_Y(y, y'), ss'])^q v^{2p} \mathrm{d}\alpha \mathrm{d}\alpha$$

$$= \int \mathcal{D}([d_X(x, x'), rr'], [d_Y(y, y'), ss'])^q \mathrm{d}\alpha \mathrm{d}\alpha$$

$$= \mathcal{H}(\alpha)$$

Both the functional and the constraint set are invariant, thus the whole CGW problem is invariant to dilations. □

The above lemma allows to normalize the plan such that one of its marginal is fixed to some value. Fixing a marginal allows to generalize the gluing lemma which is a key ingredient of the triangle inequality in optimal transport.

**Lemma 4** (Normalization lemma). *Assume there exists $\alpha \in \mathcal{U}_p(\mu, \nu)$ such that $\mathrm{CGW}(\mathcal{X}, \mathcal{Y}) = \mathcal{H}(\alpha)$. Then there exists $\tilde{\alpha}$ such that $\tilde{\alpha} \in \mathcal{U}_p(\mu, \nu)$ and $\mathrm{CGW}(\mathcal{X}, \mathcal{Y}) = \mathcal{H}(\tilde{\alpha})$ and whose marginal on $\mathfrak{C}[Y]$ is $\nu_{\mathfrak{C}[Y]} = \mathfrak{P}^{(\mathfrak{C}[Y])} \sharp \tilde{\alpha} = \delta_{\mathfrak{o}_Y} + \mathfrak{p}_\sharp(\nu \otimes \delta_1)$, where $\mathfrak{p}$ is the canonical injection from $Y \times \mathbb{R}_+$ to $\mathfrak{C}[Y]$.*

*Proof.* The proof is exactly the same as Liero et al. (2015, Lemma 7.10) and is included for completeness. Take an optimal plan $\alpha$. Because the functional and the constraints are homogeneous in $(r, s)$, the plan $\hat{\alpha} = \alpha + \delta_{\mathfrak{o}_X} \otimes \delta_{\mathfrak{o}_Y}$ verifies $\hat{\alpha} \in \mathcal{U}_p(\mu, \nu)$ and $\mathcal{H}(\hat{\alpha}) = \mathcal{H}(\alpha)$. Indeed, because of this homogeneity the contribution $\delta_{\mathfrak{o}_X} \otimes \delta_{\mathfrak{o}_Y}$ has $(r, s) = (0, 0)$ which has thus no impact.

Considering $\hat{\alpha}$ instead of $\alpha$ allows to assume without loss of generality that the transport plan charges the apex, i.e. setting

$$S = \{[x, r], [y, s] \in \mathfrak{C}[X] \times \mathfrak{C}[Y], [y, s] = \mathfrak{o}_Y\}, \tag{14}$$

one has $\omega_Y \stackrel{\mathrm{def.}}{=} \hat{\alpha}(S) \geq 1$. Then we can define the following scaling

$$v([x, r], [y, s]) = \begin{cases} s \text{ if } s > 0 \\ \omega_Y^{-1/q} \text{ otherwise.} \end{cases} \tag{15}$$

We prove now that $\mathrm{Dil}_v(\hat{\alpha})$ has the desired marginal on $\mathfrak{C}(Y)$ by considering test functions $\xi([y, s])$. We separate the integral into two parts with the set $S$, and write $\hat{\alpha} = \hat{\alpha}|_S + \hat{\alpha}|_{S^c}$ their restrictions

to $S$ and $S^c$ respectively. It reads

$$
\begin{aligned}
\int \xi([y,s]) \mathrm{dDil}_v(\hat{\alpha}) &= \int \xi([y,s/v])v^p \mathrm{d}\hat{\alpha} \\
&= \int \xi([y,s/v])v^p \mathrm{d}\,\hat{\alpha}|_S + \int \xi([y,s/v])v^p \mathrm{d}\,\hat{\alpha}|_{S^c} \\
&= \int \xi(\mathfrak{o}_Y)\omega_Y^{-1}\mathrm{d}\,\hat{\alpha}|_S + \int \xi([y,s/s])s^p \mathrm{d}\,\hat{\alpha}|_{S^c} \\
&= \xi(\mathfrak{o}_Y) \cdot \omega_Y \cdot \omega_Y^{-1} + \int \xi([y,1])s^p \mathrm{d}\hat{\alpha} \\
&= \xi(\mathfrak{o}_Y) + \int \xi(\mathfrak{p}(y,s))\mathrm{d}(\nu(y) \otimes \delta_1(s)) \\
&= \int \xi([y,s])\mathrm{d}(\delta_{\mathfrak{o}_Y} + \mathfrak{p}_\sharp(\nu \otimes \delta_1)),
\end{aligned}
$$

which is the formula of the desired marginal on $\mathfrak{C}[Y]$. Since $\hat{\alpha} \in \mathcal{U}_p(\mu,\nu)$, its dilation is also in $\mathcal{U}_p(\mu,\nu)$, and $\mathcal{H}(\alpha) = \mathcal{H}(\hat{\alpha}) = \mathcal{H}(\mathrm{Dil}_v(\hat{\alpha}))$. $\qquad\square$

### C.1.1 Proof of Theorem 1

*Non-negativity* and *symmetry* hold since $\mathcal{H}$ is a sum of non-negative symmetric terms. To prove *Definiteness*, assume $\mathrm{CGW}(\mathcal{X},\mathcal{Y}) = 0$, and write $\alpha$ an optimal plan. We have $\alpha \otimes \alpha$-a.e. that $d_X(x,x') = d_Y(y,y')$ and $rr' = ss'$ because $\mathcal{D}$ is definite (see Proposition 3). Thanks to the completeness of $(\mathcal{X},\mathcal{Y})$ and a result from Sturm (2012, Lemma 1.10), such property implies the existence of a Borel isometric bijection with Borel inverse between the supports of the measures $\psi : \mathrm{Supp}(\mu) \to \mathrm{Supp}(\nu)$, where Supp denotes the support. The bijection $\psi$ verifies $d_X(x,x') = d_Y(\psi(x),\psi(x'))$. To prove $\mathcal{X} \sim \mathcal{Y}$ it remains to prove $\psi_\sharp \mu = \nu$. Due to the density of continuous functions of the form $\xi(x)\xi(x')$, the constraints of $\mathcal{U}_p(\mu,\nu)$ are equivalent to

$$
\int_{\mathbb{R}_+} (rr')^p \mathrm{d}\alpha_1(\cdot,r)\mathrm{d}\alpha_1(\cdot,r') = \mu \otimes \mu, \qquad \int_{\mathbb{R}_+} (ss')^p \mathrm{d}\alpha_2(\cdot,s)\mathrm{d}\alpha_2(\cdot,s') = \nu \otimes \nu.
$$

Take a continuous test function $\xi$ defined on $\mathrm{Supp}(\nu)^2$. Writing $y = \psi(x)$ and $y' = \psi(x')$, one has

$$
\begin{aligned}
\int \xi(y,y')\mathrm{d}\nu\mathrm{d}\nu &= \int \xi(y,y')(ss')^p \mathrm{d}\alpha\mathrm{d}\alpha \\
&= \int \xi(\psi(x),\psi(x'))(ss')^p \mathrm{d}\alpha\mathrm{d}\alpha \\
&= \int \xi(\psi(x),\psi(x'))(rr')^p \mathrm{d}\alpha\mathrm{d}\alpha \\
&= \int \xi(\psi(x),\psi(x'))\mathrm{d}\mu\mathrm{d}\mu \\
&= \int \tilde{\xi}(x,x')\mathrm{d}\psi_\sharp\mu\mathrm{d}\psi_\sharp\mu.
\end{aligned}
$$

Since $\psi$ is a bijection, there is a bijection between continuous functions $\xi$ of $\mathrm{Supp}(\nu)^2$ and functions $\tilde{\xi}$ of $\mathrm{Supp}(\mu)^2$. Thus we obtain $\nu = \psi_\sharp\mu$ and we have $\mathcal{X} \sim \mathcal{Y}$.

It remains to prove the *triangle inequality*. Assume now that $\mathcal{D}$ satisfies it. Given three mm-spaces $(\mathcal{X},\mathcal{Y},\mathcal{Z})$ respectively equipped with measures $(\mu,\nu,\eta)$, consider $\alpha,\beta$ which are optimal plans for $\mathrm{CGW}(\mathcal{X},\mathcal{Y})$ and $\mathrm{CGW}(\mathcal{Y},\mathcal{Z})$. Using Lemma 4 to both $\alpha$ and $\beta$, we can consider measures $(\bar{\alpha},\bar{\beta})$ which are also optimal and have a common marginal $\bar{\nu}$ on $\mathfrak{C}[Y]$. Thanks to this common marginal and the separability of $(X,Y,Z)$, the standard gluing lemma (Villani, 2003, Lemma 7.6) applies and yields a glued plan $\gamma \in \mathcal{M}_+(\mathfrak{C}[X] \times \mathfrak{C}[Y] \times \mathfrak{C}[Z])$ whose respective marginals on $\mathfrak{C}[X] \times \mathfrak{C}[Y]$ and $\mathfrak{C}[Y] \times \mathfrak{C}[Z]$ are $(\bar{\alpha},\bar{\beta})$. Furthermore, the marginal $\bar{\gamma}$ of $\gamma$ on $\mathfrak{C}[X] \times \mathfrak{C}[Z]$ is in $\mathcal{U}_p(\mu,\eta)$. Indeed, $(\bar{\gamma},\bar{\alpha})$ have the same marginal on $\mathfrak{C}[X]$ and same for $(\bar{\gamma},\bar{\beta})$ on $\mathfrak{C}[Z]$, hence this property. Write

$d_X = d_X(x, x')$ for sake of conciseness (and similarly for $Y, Z$). The calculation reads

$$\text{CGW}(\mathcal{X}, \mathcal{Z})^{\frac{1}{q}} \tag{16}$$

$$\leq \left( \int \mathcal{D}([d_X, rr'], [d_Z, tt'])^q \mathrm{d}\bar{\gamma}([x, r], [z, t]) \mathrm{d}\bar{\gamma}([x', r'], [z', t']) \right)^{\frac{1}{q}} \tag{17}$$

$$\leq \left( \int \mathcal{D}([d_X, rr'], [d_Z, tt'])^q \mathrm{d}\gamma([x, r], [y, s], [z, t]) \mathrm{d}\gamma([x', r'], [y', s'], [z', t']) \right)^{\frac{1}{q}} \tag{18}$$

$$\leq \left( \int (\mathcal{D}([d_X, rr'], [d_Y, ss']) + \mathcal{D}([d_Y, ss'], [d_Z, tt']))^q \mathrm{d}\gamma \mathrm{d}\gamma \right)^{\frac{1}{q}} \tag{19}$$

$$\leq \left( \int \mathcal{D}([d_X, rr'], [d_Y, ss'])^q \mathrm{d}\gamma \mathrm{d}\gamma \right)^{\frac{1}{q}} + \left( \int \mathcal{D}([d_Y, ss'], [d_Z, tt'])^q \mathrm{d}\gamma \mathrm{d}\gamma \right)^{\frac{1}{q}} \tag{20}$$

$$\leq \left( \int \mathcal{D}([d_X, rr'], [d_Y, ss'])^q \mathrm{d}\bar{\alpha}([x, r], [y, s]) \mathrm{d}\bar{\alpha}([x', r'], [y', s']) \right)^{\frac{1}{q}}$$

$$+ \left( \int \mathcal{D}([d_Y, ss'], [d_Z, tt'])^q \mathrm{d}\bar{\beta}([y, s], [z, t]) \mathrm{d}\bar{\beta}([y', s'], [z', t']) \right)^{\frac{1}{q}} \tag{21}$$

$$\leq \text{CGW}(\mathcal{X}, \mathcal{Y})^{\frac{1}{q}} + \text{CGW}(\mathcal{Y}, \mathcal{Z})^{\frac{1}{q}}. \tag{22}$$

Since $\bar{\gamma} \in \mathcal{U}_p(\mu, \eta)$, it is thus suboptimal, which yields Equation (17). Because $\bar{\gamma}$ is the marginal of $\gamma$ we get Equation (18). Equations (19) and (20) are respectively obtained by the triangle and Minkowski inequalities, which hold because $\mathcal{D}$ which is a distance. Equation (21) is the marginalization of $\gamma$, and Equation (22) is given by the optimality of $(\bar{\alpha}, \bar{\beta})$, which ends the proof of the triangle inequality.

### C.1.2   PROOF OF THE INEQUALITY BETWEEN UGW AND CGW

The proof consists in considering an optimal plan $\pi$ for UGW, building a lift $\alpha$ of this plan into the cone such that $\mathcal{L}(\pi) \geq \mathcal{H}(\alpha)$, and prove that $\alpha$ is admissible for the program CGW, thus suboptimal.

Using Equation (8), we have

$$
\begin{aligned}
\mu \otimes \mu &= (f \otimes f)\pi_1 \otimes \pi_1 + (\mu \otimes \mu)^{\perp}, \\
(\mu \otimes \mu)^{\perp} &= \mu^{\perp} \otimes (f\pi_1) + (f\pi_1) \otimes \mu^{\perp} + \mu^{\perp} \otimes \mu^{\perp}, \\
\nu \otimes \nu &= (g \otimes g)\pi_2 \otimes \pi_2 + (\nu \otimes \nu)^{\perp}, \\
(\nu \otimes \nu)^{\perp} &= \nu^{\perp} \otimes (g\pi_2) + (g\pi_2) \otimes \nu^{\perp} + \nu^{\perp} \otimes \nu^{\perp}.
\end{aligned}
\tag{23}
$$

Recall that the canonic injection $\mathfrak{p}$ reads $\mathfrak{p}(x, r) = [x, r]$. Based on the above Lebesgue decomposition, we define the conic plan

$$\alpha = (\mathfrak{p}(x, f(x)^{\frac{1}{p}}), \mathfrak{p}(y, g(y)^{\frac{1}{p}}))_{\sharp}\pi(x, y) + \delta_{\mathfrak{o}_X} \otimes \mathfrak{p}_{\sharp}[\nu^{\perp} \otimes \delta_1] + \mathfrak{p}_{\sharp}[\mu^{\perp} \otimes \delta_1] \otimes \delta_{\mathfrak{o}_Y}. \tag{24}$$

We have that $\alpha \in \mathcal{U}_p(\mu, \nu)$. Indeed for the first marginal (and similarly for the second) we have for any test function $\xi(x)$

$$
\begin{aligned}
\int \xi(x)(r)^p \mathrm{d}\alpha &= \int \xi(x)f(x)\mathrm{d}\pi_1(x) + 0 + \int \xi(x)(1)^p \mathrm{d}\mu^{\perp}(x) \\
&= \int \xi(x)\mathrm{d}(f(x)\pi_1 + \mu^{\perp}) \\
&= \int \xi(x)\mathrm{d}\mu(x).
\end{aligned}
$$

We define $\theta^* = \theta_c^*(r, s)$ the parameter which verifies $H_c(r, s) = \theta^* L_c(r/\theta^*, s/\theta^*)$. We restrict $\alpha \otimes \alpha$ to the set $S = \{\theta_{\lambda(\Gamma)}^*((rr')^p, (ss')^p) > 0\}$. By construction, $\theta_c^*(r, s)$ is 1-homogeneous in $(r, s)$. Thus on S we necessarily have $r, r', s, s' > 0$. It yields

$$\alpha \otimes \alpha|_S = (\mathfrak{p}(x, f(x)^{\frac{1}{p}}), \mathfrak{p}(y, g(y)^{\frac{1}{p}}), \mathfrak{p}(x', f(x')^{\frac{1}{p}}), \mathfrak{p}(y', g(y')^{\frac{1}{p}}))_{\sharp}(\pi \otimes \pi).$$

Concerning the orthogonal part of the decomposition, note that whenever $\theta^* = 0$, due to the definition of $H$ the cone distance reads

$$\mathcal{D}([x, r], [y, s])^q = \varphi(0)(r^p + s^p). \tag{25}$$

It geometrically means that the shortest path between $[x, r]$ and $[y, s]$ must pass via the apex, which corresponds to a pure mass creation/destruction regime.

Furthermore we have that

$$|(\mu \otimes \mu)^\perp| = \int (r \cdot r')^p \mathrm{d} \left. (\alpha \otimes \alpha) \right|_{S^c},$$

$$|(\nu \otimes \nu)^\perp| = \int (s \cdot s')^p \mathrm{d} \left. (\alpha \otimes \alpha) \right|_{S^c}.$$

Indeed, thanks to Equation (24) we have for the first marginal that

$$|(\mu \otimes \mu)^\perp| = \left( \mu^\perp \otimes (f\pi_1) + (f\pi_1) \otimes \mu^\perp + \mu^\perp \otimes \mu^\perp \right)(X^2)$$

$$= \int (rr')^p \mathrm{d}\mathbf{p}_\sharp[\mu^\perp \otimes \delta_1] \mathrm{d}\mathbf{p}(x', f(x')^{\frac{1}{p}})_\sharp \pi_1(x')$$

$$+ \int (rr')^p \mathrm{d}\mathbf{p}(x, f(x)^{\frac{1}{p}})_\sharp \pi_1(x) \mathrm{d}\mathbf{p}_\sharp[\mu^\perp \otimes \delta_1]$$

$$+ \int (rr')^p \mathrm{d}\mathbf{p}_\sharp[\mu^\perp \otimes \delta_1] \mathrm{d}\mathbf{p}_\sharp[\mu^\perp \otimes \delta_1]$$

$$= \int (rr')^p \mathrm{d} \left. (\alpha \otimes \alpha) \right|_{S^c}.$$

Note that the last equality holds because each term of $\alpha \otimes \alpha$ involving a measure $\delta_{\circ_X}$ cancels out when integrated against $(rr')^p$.

Eventually the computation gives (thanks to Lemma 1)

$$\mathcal{L}(\pi) = \int_{X^2 \times Y^2} L_{\lambda(\Gamma)}(f \otimes f, g \otimes g) \mathrm{d}\pi \mathrm{d}\pi + \varphi(0)(|(\mu \otimes \mu)^\perp| + |(\nu \otimes \nu)^\perp|)$$

$$\geq \int H_{\lambda(\Gamma)}(f \otimes f, g \otimes g) \mathrm{d}\pi \mathrm{d}\pi + \varphi(0)(|(\mu \otimes \mu)^\perp| + |(\nu \otimes \nu)^\perp|)$$

$$\geq \int \mathcal{D}([d_X(x, x'), (f \otimes f)^{\frac{1}{p}}], [d_Y(y, y'), (g \otimes g)^{\frac{1}{p}}])^q \mathrm{d}\pi \mathrm{d}\pi$$

$$+ \int \varphi(0)(rr')^p \mathrm{d} \left. (\alpha \otimes \alpha) \right|_{S^c} + \int \varphi(0)(ss')^p \mathrm{d} \left. (\alpha \otimes \alpha) \right|_{S^c}$$

$$\geq \int \mathcal{D}([d_X(x, x'), rr'], [d_Y(y, y'), ss'])^q \mathrm{d} \left. (\alpha \otimes \alpha) \right|_{S}$$

$$+ \int \varphi(0)((rr')^p + (ss')^p) \mathrm{d} \left. (\alpha \otimes \alpha) \right|_{S^c}$$

$$\geq \int \mathcal{D}([d_X(x, x'), rr'], [d_Y(y, y'), ss'])^q \mathrm{d}\alpha \mathrm{d}\alpha$$

$$\geq \mathcal{H}(\alpha).$$

Thus we have $\mathrm{UGW}(\mathcal{X}, \mathcal{Y}) = \mathcal{L}(\pi) \geq \mathcal{H}(\alpha) \geq \mathrm{CGW}(\mathcal{X}, \mathcal{Y})$.

## D ALGORITHMIC DETAILS AND FORMULAS

### D.1 DECOMPOSITION OF KL QUADRATIC DIVERGENCE

We present in this section an additional property on the quadratic-KL divergence which allows to reduce the computational burden to evaluate it by involving the computation of a standard KL divergence.

**Proposition 9.** *For any measures $(\mu, \nu) \in \mathcal{M}_+(\mathcal{X})$, one has*

$$\mathrm{KL}(\mu \otimes \nu | \alpha \otimes \beta) = m(\nu)\mathrm{KL}(\mu|\alpha) + m(\mu)\mathrm{KL}(\nu|\beta)$$
$$+ (m(\mu) - m(\alpha))(m(\nu) - m(\beta)). \tag{26}$$

*In particular,*

$$\mathrm{KL}(\mu \otimes \mu | \nu \otimes \nu) = 2m(\mu)\mathrm{KL}(\mu|\nu) + (m(\mu) - m(\nu))^2. \tag{27}$$

*Proof.* Assuming $\mathrm{KL}(\mu \otimes \nu | \alpha \otimes \beta)$ to be finite, one has $\mu = f\alpha$ and $\nu = g\beta$. It reads

$$\mathrm{KL}(\mu \otimes \nu | \alpha \otimes \beta) = \int \log(f \otimes g)\mathrm{d}\mu\mathrm{d}\nu - m(\mu)m(\nu) + m(\alpha)m(\beta)$$

$$= m(\nu)\int \log(f)\mathrm{d}\mu + m(\mu)\int \log(g)\mathrm{d}\nu$$
$$- m(\mu)m(\nu) + m(\alpha)m(\beta)$$
$$= m(\nu)\big[\mathrm{KL}(\mu|\alpha) + m(\mu) - m(\alpha)\big]$$
$$+ m(\mu)\big[\mathrm{KL}(\nu|\beta) + m(\nu) - m(\beta)\big]$$
$$- m(\mu)m(\nu) + m(\alpha)m(\beta)$$
$$= m(\nu)\mathrm{KL}(\mu|\alpha) + m(\mu)\mathrm{KL}(\nu|\beta)$$
$$+ m(\mu)m(\nu) - m(\nu)m(\alpha) - m(\mu)m(\beta) + m(\alpha)m(\beta)$$
$$= m(\nu)\mathrm{KL}(\mu|\alpha) + m(\mu)\mathrm{KL}(\nu|\beta)$$
$$+ (m(\mu) - m(\alpha))(m(\nu) - m(\beta)).$$

$\square$

We now prove Proposition 4 which applies the above result.

**Proposition 10.** *For a fixed $\gamma$, the optimal $\pi \in \arg\min_\pi \mathcal{F}(\pi, \gamma) + \varepsilon\mathrm{KL}(\pi \otimes \gamma | (\mu \otimes \nu)^{\otimes 2})$ is the solution of $\min_\pi \int c_\gamma^\varepsilon(x, y)\mathrm{d}\pi(x, y) + \rho m(\gamma)\mathrm{KL}(\pi_1|\mu) + \rho m(\gamma)\mathrm{KL}(\pi_2|\nu) + \varepsilon m(\gamma)\mathrm{KL}(\pi|\mu \otimes \nu)$, where $m(\gamma) \stackrel{\text{def.}}{=} \gamma(X \times Y)$ is the total mass of $\gamma$, and where we define the cost and weight associated to $\gamma$ as*

$$c_\gamma^\varepsilon(x, y) \stackrel{\text{def.}}{=} \int \lambda(\Gamma(x, \cdot, y, \cdot))\mathrm{d}\gamma + \rho \int \log(\frac{\mathrm{d}\gamma_1}{\mathrm{d}\mu})\mathrm{d}\gamma_1 + \rho \int \log(\frac{\mathrm{d}\gamma_2}{\mathrm{d}\nu})\mathrm{d}\gamma_2 + \varepsilon \int \log(\frac{\mathrm{d}\gamma}{\mathrm{d}\mu\mathrm{d}\nu})\mathrm{d}\gamma.$$

*Proof.* First note that $\mathcal{F}(\gamma, \pi) = \mathcal{F}(\pi, \gamma)$ so that minimizing with the first or the second argument gives the same solution. The rest follows from the relation

$$\mathrm{KL}(\pi_1 \otimes \gamma_1 | \mu \otimes \mu) = m(\gamma)\mathrm{KL}(\pi_1|\mu) + m(\pi)\mathrm{KL}(\gamma_1|\mu) + (m(\gamma) - m(\mu))(m(\pi) - m(\mu)),$$

and also from $\mathrm{KL}(\pi_1|\mu) = \int \log(\frac{\mathrm{d}\gamma_1}{\mathrm{d}\mu})\mathrm{d}\gamma_1 - (m(\gamma) - m(\mu))$. Similar formulas hold for $(\pi_2, \gamma_2)$ and $(\pi, \gamma)$. $\square$

## D.2 Discrete setting and formulas

In order to implement those algorithms, one consider discrete mm-spaces $X = (x_i)_{i=1}^n$ and $Y = (y_j)_{j=1}^m$, endowed with discrete measures $\mu = \sum_i \mu_i \delta_{x_i}$ and $\nu = \sum_j \nu_j \delta_{y_j}$, where $\mu_i, \nu_j \geq 0$. The distance matrices are $D_{i,i'}^X \stackrel{\text{def.}}{=} d_X(x_i, x_{i'})$ and $D_{j,j'}^X \stackrel{\text{def.}}{=} d_X(y_j, y_{j'})$. Transport plan are thus also discrete $\pi = \sum_{i,j} \pi_{i,j}\delta_{(x_i, y_j)}$.

The functional $\mathcal{L}$ now reads in this discrete setting

$$\int (d_X(x, x') - d_Y(y, y'))^2\mathrm{d}\pi(x, y)\mathrm{d}\pi(x', y') = \sum_{i,j,k,\ell} (D_{i,j}^X - D_{k,\ell}^Y)^2\pi_{i,k}\pi_{j,\ell},$$

$$\text{and}\quad \text{KL}(\pi_1 \otimes \pi_1 | \mu \otimes \mu) = \sum_{i,j} \log\left(\frac{\pi_{1,i}\pi_{1,j}}{\mu_i\mu_j}\right)\pi_{1,i}\pi_{1,j} - \sum_{i,j}\pi_{1,i}\pi_{1,j} + \sum_{i,j}\mu_i\mu_j$$

$$= 2m(\pi)\sum_i \log\left(\frac{\pi_{1,i}}{\mu_i}\right)\pi_{1,i} - m(\pi)^2 + m(\mu)^2,$$

where we define the marginals $\pi_{1,k} \overset{\text{def.}}{=} \sum_j \pi_{k,j}$, $\pi_{2,\ell} \overset{\text{def.}}{=} \sum_i \pi_{i,\ell}$ and $m(\pi) = \sum_{i,j}\pi_{i,j}$.

When one runs the stabilized implementation of Sinkhorn's iterations with a ground cost $C_{i,j} = C(x_i, y_j)$ between the points, it is necessary to use a Log-Sum-Exp reduction which reads

$$f_i \leftarrow -\frac{\varepsilon\rho}{\varepsilon + \rho}\text{LSE}_j\big[(g_j - C_{i,j})/\varepsilon + \log(\mu_j)\big] \tag{28}$$

where $\text{LSE}_j$ is a reduction performed on the index $j$. It reads

$$\text{LSE}_j(C_{i,j}) \overset{\text{def.}}{=} \log\left(\sum_j \exp(C_{i,j} - \max_k C_{i,k})\right) + \max_k C_{i,k}, \tag{29}$$

where the logarithm and exponential are pointwise operations.

---

**Algorithm 2 – UGW$(\mathcal{X}, \mathcal{Y}, \rho, \varepsilon)$** in discrete form

---

**Input:** mm-spaces $\mathcal{X} = (D_{i,j}^X, (\mu_i)_i)$ and $\mathcal{Y} = (D_{i,j}^Y, (\nu_j)_j)$, relaxation $\rho$, regularization $\varepsilon$
**Output:** approximation $(\pi, \gamma)$ minimizing 6

1: Initialize matrix $\pi_{i,j} = \gamma_{i,j} = \mu_i\nu_j/\sqrt{(\sum_i \mu_i)(\sum_j \nu_j)}$, vector $g_j^{(s=0)} = 0$.
2: **while** $\pi$ has not converged **do**
3:      Update $\pi \leftarrow \gamma$
4:      Define $m(\pi) \leftarrow \sum_{i,j}\pi_{i,j}$, $\tilde\rho \leftarrow m(\pi)\rho$, $\tilde\varepsilon \leftarrow m(\pi)\varepsilon$
5:      Define $c \leftarrow \text{ComputeCost}(\mathcal{X}, \mathcal{Y}, \pi, \rho, \varepsilon)$
6:      **while** $(f, g)$ has not converged **do**
7:          $f \leftarrow -\frac{\tilde\varepsilon\tilde\rho}{\tilde\varepsilon+\tilde\rho}\log\left[\sum_j \exp\left((g_j - c_{i,j})/\tilde\varepsilon + \log\nu_j\right)\right]$
8:          $g \leftarrow -\frac{\tilde\varepsilon\tilde\rho}{\tilde\varepsilon+\tilde\rho}\log\left[\sum_i \exp\left((f_i - c_{i,j})/\tilde\varepsilon + \log\mu_i\right)\right]$
9:      Update $\gamma_{i,j} \leftarrow \exp\left[(f_i + g_j - c_{i,j})/\tilde\varepsilon\right]\mu_i\nu_j$
10:     Rescale $\gamma \leftarrow \sqrt{m(\pi)/m(\gamma)}\gamma$
11: Return $(\pi, \gamma)$.

---

We also provide an algorithm that computes the cost $c_\pi^\varepsilon$ defined in Proposition (10). We focus on the case $D_\varphi = \rho\text{KL}$ and $\lambda(t) = t^2$ which is computable with complexity $O(n^3)$ as shown in Peyré et al. (2016). Indeed, note that one has

$$\int (d_X(x, x') - d_Y(y, y'))^2 d\pi(x', y') = \int d_X(x, x')^2 d\pi_1(x') + \int d_Y(y, y')^2 d\pi_2(y')$$

$$- 2\int d_X(x, x')d_Y(y, y')d\pi(x', y').$$

---

**Algorithm 3 – ComputeCost$(\mathcal{X}, \mathcal{Y}, \pi, \rho, \varepsilon)$** in discrete form

---

**Input:** mm-spaces $\mathcal{X} = (D_{i,j}^X, (\mu_i)_i)$ and $\mathcal{Y} = (D_{k,\ell}^Y, (\nu_j)_j)$, transport matrix $(\pi_{j,k})_{j,k}$, relaxation $\rho$, regularization $\varepsilon$
**Output:** cost $c_\pi^\varepsilon$ defined in Proposition 10

1: Compute $\pi_{1,j} \leftarrow \sum_k \pi_{j,k}$ and $\pi_{2,k} \leftarrow \sum_j \pi_{j,k}$          $\triangleright\ \pi_1 = \pi\mathbf{1}$ and $\pi_2 = \pi^\top\mathbf{1}$
2: Compute $A_i \leftarrow \sum_j (D_{i,j}^X)^2 \pi_{1,j}$                            $\triangleright\ A = (D^X)^{\circ 2}\pi_1$
3: Compute $B_\ell \leftarrow \sum_k (D_{k,\ell}^Y)^2 \pi_{2,k}$                         $\triangleright\ B = (D^Y)^{\circ 2}\pi_2$
4: Compute $C_{i,\ell} \leftarrow \sum_j D_{i,j}^X\left(\sum_k D_{k,\ell}^Y \pi_{j,k}\right)$              $\triangleright\ C = D^X\pi D^Y$
5: Compute $E \leftarrow \rho\sum_j \log\left(\frac{\pi_{1,j}}{\mu_j}\right)\pi_{1,j} + \rho\sum_k \log\left(\frac{\pi_{2,k}}{\nu_k}\right)\pi_{2,k} + \varepsilon\sum_{j,k}\log\left(\frac{\pi_{jk}}{\mu_j\nu_k}\right)\pi_{j,k}$
6: Return $c_{\pi,i,\ell}^\varepsilon \leftarrow A_i + B_\ell - 2C_{i,\ell} + E$

---

