# OpenReview forum: "The Unbalanced Gromov Wasserstein Distance: Conic Formulation and Relaxation"
_ICLR.cc/2021/Conference — Reject_

### Official Review · AnonReviewer2 · 2020-10-27
**Nice theoretical results, poor experimental Section**

**Rating:** 6
**Confidence:** 4

**Review:**

The paper introduces a novel unbalanced Gromov-Wasserstein type problem. The Gromov-Wasserstein distance is very useful in practice for comparing probability distributions that do not lie in the same metric spaces. It has recently found several successful applications in ML for computational chemistry, graphs comparisons or NLP. Following previous works on unbalanced optimal transport (i.e. soft constraints over marginals enforcement of the coupling matrix), and the rationale that disposing of unbalanced versions of transport problems can alleviate in some ways presence of outliers or noise in the distributions, the authors propose two ‘unbalanced’ variants of the Gromov-Wasserstein (GW) problem, that allow comparison of metric spaces with arbitrary positive measures up to isometries (I.e. rigid transformations).

The paper is fairly well written, original, and the related works is particularly complete. The theoretical part of the paper is sound, rigorous and well-motivated, and I learnt many things from it. The idea of using a quadratic \phi-divergence is neat and particularly clever. Yet, the paper requires, to some extent, very good notions in (unbalanced) optimal transport and probability, and I wonder, if some notations could have been eased a little bit (for instance, it seems that D_\phi is usually chosen as KL), but I guess the choice was made to be as general as possible.  While I did not check in details all the proofs in the appendix Section, I believe the work is solid. The algorithm parts is a little less satisfactory, as it amounts to optimize a upper bound  of the described problem, following existing work from [Mémoli11]. Though appealing, this upper bound is known to lose some properties of the original GW distance (for instance, two close points in the source can be matched to two distant points in the target if they share a very similar ‘distance profile’ or distance distributions wrt. the other samples). It seems that this effect is observable in Figure 2, where parts of a moon are ‘flipped’ in the matching. Combined with entropy and unbalanced formulation, I guess the final result can be very hard to interpret in a practical setting.
Finally, the weakest part of the paper is the experimental Section. Only two toy examples are presented. While the method could have been used in many settings (graph classification, embedding matching in NLP or  even graph matching, for which many algorithms exist, etc.), it is very hard to conclude about the practical interest of the method.
While I guess this is not a problem if one focuses on the original contribution of this novel unbalanced distance, it is more problematic, in the reviewers perception, for a machine learning venue such as ICLR.

For this reason, I will solely suggest a ‘weak accept’ decision, while I really believe the theoretical sections have a lot of merits.

Minor comments.

- On page 1, it is said that « the paper defines for the first time a class of distances between the [aforementioned] objects ». This claim is a bit strong knowing that GW has already been used in several (cited) applications. I guess authors could be a little bit more precise on the meaning of this sentence;
- Page 4, just after definition 1, it seems (at least with my PDF reader), that there is a problem in the symbol used for \phi on the last line of the paragraph;
- The claim that a cost of O(n^3) in time and O(n^2) in memory allows to scale to large problems is a bit strong. In practice, what is the maximum size of the problem that can be addressed in reasonable time with this method ? I expect that handling graphs that have more than 10k nodes for instance in very difficult.
- The final rescaling of \gamma (line 8 of Algorithm 1), is a little bit difficult to understand from the alternating minimization point of view.
- In the experimental Section, which algorithm is used to compute GW ? Also, I guess Figure 3 could present the original graphs (without scaling the dots), to gain a better understanding of the original problem


EDIT after rebuttal period
---------------------------------------
many thanks to the authors for taking into consideration my comments. I decided not to change my note because I still believe the paper lacks of a significative experimental Section.

---

> ### Author Response · Authors · 2020-11-15
> **Personnal answer to reviewer 2**
>
> We would like to thank the reviewer for his careful reading of our submission. We wrote a common answer addressed to all reviewers in a comment above, and provide here a personal answer to the reviewer’s comments.
>
> - *“I wonder, if some notations could have been eased a little bit”:*
> We acknowledge that the generality of the approach makes the paper notation-heavy. However, the degree of freedom in choosing general divergences makes the method versatile and will likely be useful for future applications. To showcase the difference with KL, we will add in the revised version a simulation using TV or the method of [Chapel et al.] which is connected to our TV setting.
> - *“The algorithm parts is a little less satisfactory, as it amounts to optimize an upper bound of the described problem, following existing work from [Mémoli11].”:*
> First note that we do not solve for an upper bound, but a lower bound when introducing a pair ($\gamma$,$\pi$). This approach is the de-facto standard approach to solve this type of bilinear problem. A theoretical justification is provided in [1] which proves that when the functional is concave (which is the case for GW on Euclidean space) then the bound is tight. Extending this to general cases is unclear, but in practice, we found that the algorithm always converges to $\gamma$=$\pi$, thus showing that the bound is tight. We will further explain this in the revised version.
> - *“It seems that this effect is observable in Figure 2, where parts of a moon are ‘flipped’ in the matching.”:*
> in this specific case, the  flip is not due to a suboptimal matching, but rather to local symmetries of the pointcloud (so that the loss function has several almost global minimizers).
> - *“The claim that a cost of O(n^3) in time and O(n^2) in memory allows to scale to large problems is a bit strong”:*
> Indeed, we will lower our claim, our method only scales to medium size problems (~10K). Extending this to larger problems would require using kernel compression methods such as hierarchical factorization [2] or Nystrom approximations of the cost [3]. We will mention this, but left the study of these approximation methods to future works.
> - *“The final rescaling of $\gamma$ (line 8 of Algorithm 1), is a little bit difficult to understand from the alternating minimization point of view”:*
> This is only to remove the invariance of the problem by rescaling ($\pi$,$\gamma$)->($\pi$ *c, $\gamma$ /c), which is important to ensure that one hopefully gets $\pi$=$\gamma$ at convergence (which is observed in practice), but up to fixing this invariance it has no impact.
> - *“In the experimental Section, which algorithm is used to compute GW ?”:*
> We use the implementation of GW available in the POT library, which is equivalent to our algorithm when using $\epsilon$=0 (no entropy regularization) and no mass relaxation (balanced case).
> - *“Also, I guess Figure 3 could present the original graphs (without scaling the dots), to gain a better understanding of the original problem”:*
> The graph plotted for GW in Figure 3 should be taken as reference since there is no rescaling due to mass creation.
>
> References:
>
> [1] Konno, H. (1976). Maximization of a convex quadratic function under linear constraints.
>
> [2] Altschuler, J., Bach, F., Rudi, A., & Niles-Weed, J. (2019). Massively scalable Sinkhorn distances via the Nyström method.
>
> [3] Xu, H., Luo, D., & Carin, L. (2019). Scalable Gromov-Wasserstein learning for graph partitioning and matching.

---

### Official Review · AnonReviewer3 · 2020-10-27
**The paper is quite interesting but lacks novelty.**

**Rating:** 5
**Confidence:** 5

**Review:**

In the paper, the authors propose two versions of Gromov-Wasserstein distance when the weights of measures do not need to sum up to one. The first version, named Unbalanced Gromov-Wasserstein (UGW), is a direct application of unbalanced optimal transport (UOT) to the setting of Gromov-Wassertstein. The second version, named Conic Gromov-Wasserstein (CGW), is an extension of the conic formulation of UOT to the setting of Gromov-Wasserstein. The authors also show that CGW is a distance in the metric-measure spaces and is an lower bound of the UGW. Finally, the authors also provide some experiments with their proposed divergences.

In my opinion, the two proposed versions of Gromov-Wasserstein are quite interesting but lack novelty. They are simply extensions of the framework of UOT to the framework of Gromov-Wasserstein. Here are my other comments with the paper:

(1) The CGW is introduced as a lower bound of UGW; however, it is computationally expensive and practically out-of-reach as the authors admitted in the paper. Therefore, I am a bit confused with the proposal of CGW in the paper as it is not useful in practice. The theory of CGW also does not appear deep enough for me to appreciate the contribution. Unless the authors can provide some thoughts on how to practically implement CGW in the rebuttal, I feel that the inclusion of CGW is quite weak and does not add extra practical value to the paper.

(2) Going back to the UGW, it is a direct extension of the framework from UOT to that of Gromov-Wasserstein. The theories (Proposition 1,2 and Lemma 1) are fine and not hard to obtain given those from UOT. In the revision, I think the authors should give some examples of $\varphi$ satisfying the assumptions in Propositions 1 and 2 to help the readers to understand these results better. In Lemma 1, the authors should provide some intuition about $L_{c}(a, b)$ as it is a key term used later to define the conic discrepancies $\mathcal{D}$ in the definition of CGW.

(3) There is an interesting algorithmic perspective of using unbalanced version over the balanced version. For the setting of OT, the recent paper of [1] shows that solving the UOT (by Sinkhorn algorithm) when the divergence is KL has the complexity of $n^2/ \varepsilon$, which is near optimal, where $\varepsilon$ denotes the desired accuracy to approximate the UOT by the entropic UOT. On the other hand, the complexity of using Sinkhorn algorithm for solving the OT is at the order of $n^2/ \varepsilon^2$. Therefore, by considering the relaxation over constraints, solving UOT is in fact algorithmically favorable than that of solving OT.

Going back to the UGW, the relaxation of constraints, in my opinion, may also help the convergence of algorithm (specifically Algorithm 1 in the paper) comparing to that of the GW (via Sinkhorn algorithm) though the rigorous theoretical analyses of these algorithms can be very tricky in general. However, I think the authors also can provide some numerical experiments to see whether this phenomenon holds true.

(4) The literature of using UOT as a robust version of OT in practical applications of deep generative models and domain adaptation has been considered recently; see for example the paper [2]. The authors may consider adding this reference in the related works and contributions.


References:

[1] K Pham, K Le, N Ho, T Pham, H Bui. On Unbalanced Optimal Transport: An Analysis of Sinkhorn Algorithm. ICML, 2020

[2] Y. Balaji, R. Chellappa, S. Feizi. Robust Optimal Transport with Applications in Generative Modeling and Domain Adaptation. NeurIPS, 2020

---

> ### Author Response · Authors · 2020-11-15
> **Personnal answer to reviewer 3**
>
> We would like to thank the reviewer for the careful reading of our submission. We wrote a common answer addressed to all reviewers in a comment above, and provide here a personal answer to the reviewer’s comments.
>
> - *“They are simply extensions of the framework of UOT to the framework of Gromov-Wasserstein. “:*
> We respectfully disagree with the reviewer on this point. In the “common answers” we explain why we believe our contributions are significant both with respect to previous attempts to address unbalanced GW and with respect to unbalanced OT.
> - *“CGW is quite weak and does not add extra practical value”:*
> We disagree with this statement.  [De Ponti et al.] and our paper appeared at the same time and are the only valid construction of unbalanced distances. None are implementable, but our is controlled (upper-bounded) by a valid divergence.
> - *“ [...] the authors can provide some thoughts on how to practically implement CGW in the rebuttal [...]” : *
> Unfortunately, the conic formulation seems much more challenging to solve, in particular it cannot be cast as a finite dimensional optimization. A workaround is to approximate the lifted measure using a particle system. We will mention this in an updated version of the paper, but will leave these extensions for future works.
> - *“The theories (Proposition 1,2 and Lemma 1) are fine and not hard to obtain given those from UOT”:*
> We respectfully disagree with the reviewer on this point. We provided in the common answers some illustrative examples showcasing the difference in the proof of UOT and UGW theoretical results.
> - *“In the revision, I think the authors should give some examples of satisfying the assumptions in Propositions 1 and 2 to help the readers to understand these results better.”:*
> The assumptions are satisfied for all common entropies such as KL and TV. The compactness assumption holds in cases of practical interest, when working on bounded (and in particular finite) sets (e.g. graphs, pointclouds). We will add more details on this in a revised manuscript.
> - *“In Lemma 1, the authors should provide some intuition about $L_c$ as it is a key term used later to define the conic discrepancies D in the definition of CGW”:*
> We acknowledge that most of the information regarding $L_c$ and $H_c$ are buried in the supplementary material. $L_c$ appears naturally when re-writing the initial optimization problem using the inverse of the marginal densities (denoted f and g) which itself is useful to separate the pure transport of mass (measured by $L_c$) and the pure creation/destruction (measured by the term multiplied by $\psi_\infty$ in (4)). $H_c$ is the 1-homogeneous transformation of $L_c$, which is required in order to perform a conic lifting of the problem. We will update the final version of the manuscript to include additional intuition about these functionals.
> - *“There is an interesting algorithmic perspective of using unbalanced version over the balanced version. “*
> We thank the reviewer for pointing this to us. This provides an additional motivation (convergence speed) to use unbalanced methods in GW.  We will add this reference in the paper.
> - *“The literature of using UOT as a robust version of OT in practical applications of deep generative models and domain adaptation has been considered recently”:*
> We thank the reviewer for this very relevant reference which will be added to the paper.

---

> > ### Comment · AnonReviewer3 · 2020-11-24
> > **Thanks for the response**
> >
> > I would like to thank the authors for spending time responding to my comments.
> >
> > Frankly speaking, I am still not convinced about the introduction of CGW and in fact get confused about its point in the paper. If CGW is expensive and hard to implement, I still do not understand why I should care about it (though I agree that it seems to be quite interesting from the theoretical perspective). Furthermore, as other reviewers pointed out, the current experiments are quite limited. I feel that the paper still needs more works in terms of experiments to fully explore the potential of unbalanced Gromov-Wasserstein divergence in machine learning and deep learning applications.
> >
> > For the above reason, I decide to keep my current score with the paper.

---

### Official Review · AnonReviewer1 · 2020-10-28
**The Unbalanced Gromov Wasserstein Distance: Conic Formulation and Relaxation**

**Rating:** 7
**Confidence:** 4

**Review:**

The authors consider the Gromov Wasserstein (GW) problem for metric measure spaces having different masses (i.e., Unbalanced GW). Similar to the ideas of unbalanced optimal transport (UOT), they proposed to use a quadratic divergence to relax the marginal constraints (instead of divergence as in UOT). When divergence is KL, the authors derive a GPU-friendly algorithm for the UGW  which relies on the unbalanced Sinkhorn algorithm. Additionally, the authors also propose a variant of UGW, namely Conic Gromov-Wasserstein (CGW) to address the different masses of metric measure spaces. The authors propose that CGW has nice properties (Theorem 1). However, there is no algorithm to solve the CGW yet.

+ It is easy to follow the paper. The authors provide rigorous details for the unbalanced GW.

+ For the UGW with divergence, it seems that it is not surprised for this idea by extending the ideas of entropic OT for unbalanced OT into entropic GW for unbalanced GW (e.g., using Sinkhorn iterations).

+ The proposed conic GW is theoretically interesting since it comes up with no algorithm yet. The authors also draw a connection between CGW and UGW.

+ It seems that the experiments are quite simple (with some toy examples). It seems better if the authors use UGW in some applications (e.g., some motivating applications for the unbalance case).

Some of my other concerns are as follows:
+ The authors propose to use quadratic divergence in UGW. Could the authors give more motivation/explanation about this approach, why not just divergence as in unbalanced OT (Although the GW is a quadratic assignment, the unbalanced problem for GW comes from the marginal constraints, why not just simply use the divergence between the marginal and measures)?
+ The authors proved nice properties for the proposed CGW and draw its connection to UGW. However, CGW has no algorithm yet. I wonder whether there exist some special instances of UGW (with some specific divergence, e.g., total variation or KL), one may have some interesting properties as in CGW?

Overall, I lean on the positive side. (i) The ideas of UGW may not be a surprise (together with its algorithm). (ii) The proposed CGW has good properties for GW problems for metric measure spaces having different masses, however, there is no efficient algorithm for it yet as (a trade-off). (iii) It seems better in case the authors improve experiments (e.g., use UGW in some motivating applications for the unbalanced case).

---

> ### Author Response · Authors · 2020-11-15
> **Personnal answer to Reviewer 1**
>
> We would like to thank the reviewer for the careful reading of our submission. We wrote a common answer addressed to all reviewers in a comment above, and provide here a personal answer to the reviewer’s comments.
>
> - *“It seems better if the authors use UGW in some applications”:*
> as detailed in the common answers, our goal in the numerical simulations was rather to give some intuitions about the relevance and impact of mass relaxation.
> - *“Could the authors give more motivation/explanation about this approach, why not just divergence as in unbalanced OT”:*
> as detailed in the common answers, using “classical” divergences makes little sense because losing mass-homogeneity would lead to degenerate loss functions for measures with very small or very large total mass.
> - *“The proposed conic GW is theoretically interesting since it comes up with no algorithm yet.[...]  However, CGW has no algorithm yet.”:*
> Unfortunately, the conic formulation seems much more challenging to solve, in particular it cannot be cast as a finite dimensional optimization. A workaround is to approximate the lifted measure using a particle system. We will mention this in an updated version of the paper, but will leave these extensions for future works.
> - *“I wonder whether there exist some special instances of UGW (with some specific divergence, e.g., total variation or KL), one may have some interesting properties as in CGW?”:*
> as detailed in the common answers, this is a very relevant but difficult question, which we leave open for future works, and will mention in the perspectives of the paper.

---

> > ### Comment · AnonReviewer1 · 2020-11-24
> > **Thanks for your clarification**
> >
> > Thank you for your clarification.
> > I am happy to increase my score (6 --> 7).

---

### Official Review · AnonReviewer4 · 2020-10-28
**A nice theoretical contribution that extends the Gromov-Wasserstein distance to the unbalanced setting, with limited experimental validation**

**Rating:** 6
**Confidence:** 4

**Review:**

Summary and overall recommendation:

This paper introduces two generalizations of the Gromov-Wasserstein distance to the case where the measures are unbalanced (i.e., not necessarily probability measures). While one of these is a proper distance, it is computationally infeasible. The other one, an upper bound relaxation, is not a proper distance but is much more computationally feasible. The main contribution of the paper is to prove some fundamental properties of these objects: positivity and definiteness. The paper proposes an algorithm for solving the upper-bound version, and shows proof-of-point experiments on very simple 2D settings.

Despite some shortcomings in novelty, experimental evaluation, and presentation detailed below, I find the theoretical contribution of this paper to be just enough to carry it, so I'm (weakly) recommending acceptance.

Strengths:

1. Albeit relying on ideas that are already quite popular in the OT literature (Gromov-Wasserstein distance, Unbalanced OT), this paper manages to weave them together in a very compelling way
2. The paper has sound theoretical footing, and a makes a solid theoretical contribution.
3. The writing and exposition are mostly clear, intuitive and well developed

Weaknesses:
1. The flip side of strength (1) is that the novelty of this paper is arguably limited, considering it builds on known techinques, and is addressing a problem that has been tackled in at least two recent works (De Ponti & Mondino 2020, Chapel et al. 2020).
2. Some of the assumptions and results are not discussed in enough detail
3. The experimental section is limited to very simple settings, that either conisder measures on the same space (defeating the purpose of GW) (Fig 1,2), or do so for very simple synthetic data (Fig3). In particular, none of the motivating applications mentioned in the introduction (NLP, chemistry, graph matching).

Other comments:
1. It would be useful if the paper discussed the assumptions that go into Prop 1 (superlinearity, compact level sets) in more detail. What families of $\phi$-divergences satisfy these?
2. I understand that Lemma 1 is a tool to draw a connection between the two approaches, but I find its introduction dry and abrupt. It would help to provide some discussion on the intuition on introducing such a perspective transform.
3. The presentation of the conic formulation needs fmore hand-holding. What's the motivation for this cumbersome the conic formulation? Are there other prior examples of such conic distances (even if not on mm spaces)?
4. I would have appreciated a discussion on the tightness of the bound of Theorem 1. Without this, the purported possibility to use one as a computationally feasible drop-in for the other is not fully supported.
5. I don't think $\gamma_1$ and $\gamma_2$ in page 6 were formally defined anywhere
6. There is no comparison to other unbalanced methods (either OT or GW) except for one setting in Fig 3. I suspect most of these alternatives would produce similar results in Fig 1, so what is the competitive advantage of this method? If it is complexity/runtime (vs other unbalanced GW methods), then there should be either a formal complexity analysis, or a empirical runtime comparison. Without these, the paper feels incomplete.

Questions:
1. What is meant by the last sentence in the first paragraph? This reads to be as if this paper claims to propose the first distance between the objects tackled in those works, which is certainly not the case for all of those which use GW or variations of the Wasserstein distance.
2. Is there a reference for the quadratic $\phi$ divergences introduced in 2.1?
3. Do any of the experiments use the debiased UGW mentioned in page 7?

Minor comments and typos:
* pg 3 shed some lights
* missing "is" in first line of Prop 1
* pg 8 "close to be isometric"

---

> ### Author Response · Authors · 2020-11-15
> **Personnal answer to Reviewer 4**
>
> We would like to thank the reviewer for the careful reading of our submission. We wrote a common answer addressed to all reviewers in a comment above, and provide here a personal answer to the reviewer’s comments.
>
> Weaknesses:
> - *“The novelty of this paper is arguably limited”:*
> We respectfully disagree with such a statement, and respond to this in the common answers.
> - *“The experimental section is limited to very simple settings”:*
> We refer the reviewer to the common answers for a response to this remark.
>
> Other comments:
> - *“It would be useful if the paper discussed the assumptions that go into Prop 1”:*
> They are satisfied for most common f-divergences (TV, KL) and with bounded sets (e.g. point clouds or graphs). We will clarify this in the paper.
> - *“I understand that Lemma 1 is a tool to draw a connection between the two approaches, but I find its introduction dry and abrupt.”:*
> We agree it is abrupt at first, and we will update the paper to improve its understanding. We had to trade brevity for clarity, but left details available in appendix A. The role of Lemma 1 is to introduce the cost $L_c$ which separates the pure creation penalty and includes the rest (partial transport, partial creation) in a new ‘transport’ cost.
> - *“The presentation of the conic formulation needs more hand-holding. What's the motivation for this cumbersome the conic formulation?”:*
> The goal is to express variational unbalanced problems as classical (balanced problems) over a lifted conic space. Similarly to the unbalanced OT framework (although the problem proofs are different, as detailed in the common answers), this lifting is pivotal to prove distance properties.
> - *“I would have appreciated a discussion on the tightness of the bound of Theorem 1.”:*
> As detailed in the common answers, it is an open question we wish to solve in future works.
> - *“I don't think $\gamma_1$ and $\gamma_2$ in page 6 were formally defined anywhere”:*
> This corresponds to the marginals of a transport plan $\gamma$, which is defined below Equation (1). We will recall this in the final version.
> - *“The experimental section is limited to very simple settings, that either consider measures on the same space (defeating the purpose of GW) “:*
> We believe that despite its apparent simplicity, GW over Euclidean spaces is arguably the one which is the most used in practice (for e.g. shape matching in imaging [1] or word embeddings comparison in NLP  [2, 3]).
> - *“There is no comparison to other unbalanced methods (either OT or GW) except for one setting in Fig 3. I suspect most of these alternatives would produce similar results in Fig 1, so what is the competitive advantage of this method? “:*
> We compare UGW and UOT in Figure 2, to highlight the isometry invariance of UGW. The comparison of UGW with GW highlights the ability to deal with outliers (Fig 2) and unbalanced masses (Fig 1). We acknowledge that our simulations lack a comparison with other divergences, and in particular TV (and [Chapel et al.] which also makes use of a TV fidelity), which is expected (similar to UOT) to produce significantly different results. We will add these simulations in the final version of the paper.
> - *“there should be either a formal complexity analysis, or an empirical runtime comparison.“:*
> Our only competitor is [Chapel et al.], which uses a Frank-Wolfe algorithm, so we believe the comparison would be unfair since F-W does not run on GPU and does not offer an epsilon-approximation mechanism. Besides partial GW is a different optimization problem, making the comparison difficult. In the revised version, we will show a visual comparison of the matchings obtained by partial GW and UGW (sharp vs. soft transitions).
>
> Questions:
> - *“What is meant by the last sentence in the first paragraph?”:*
> We will reformulate to explain (as detailed in the common answers) that our formulation is the first which yields both theoretical results and a computable algorithm at the same time. Note that the distance in [De Ponti et al.] appeared online at the same time as our submission, and is not amenable to a finite dimensional optimization problem when considering finite spaces.
> - *“Is there a reference for the quadratic $\phi$ divergences introduced in 2.1?”:*
> To the best of our knowledge we found no prior reference using such quadratic divergence.
> - *“Do any of the experiments use the debiased UGW mentioned in page 7?”:*
> In all the experiments we compute UGW for a small regularization parameter epsilon, so that debiasing is not required. We will move this idea in the perspective section to avoid confusion.
>
> References:
>
> [1] Rodola et al., A game-theoretic approach to deformable shape matching.
>
> [2] Alvarez-Melis et al., Gromov-wasserstein alignment of word embedding spaces.
>
> [3] Grave et al., Unsupervised align-ment of embeddings with wasserstein procrustes

---

> > ### Comment · AnonReviewer4 · 2020-11-24
> > **Thank you for the answers**
> >
> > I thank the authors for the detailed answers to my questions. They confirm my appreciation of the theoretical contributions of this paper and my recommendation for acceptance, but given the (still mostly unaddressed) limitations in the experimental evaluation, I am inclined to maintain my current score.

---

### Author Response · Authors · 2020-11-15
**Common answers to the reviewers**

We would like to thank all the reviewers for their careful reading of our manuscript. We would like to start by addressing some common remarks made by the reviewers.

**Novelty with respect to previous works:**
Only 2 previous works address the computation of unbalanced GW : (i) [Chapel et al.] precedes ours but relies on a Frank-Wolfe solver which is not easily parallelizable on GPUs ; furthermore, it does not define a distance ; (ii)  [De Ponti et al.] appeared on arxiv concomitantly to this submission and also defines a distance as our conic formulation, but is not amenable to a finite dimensional optimization problem when considering finite spaces (does not come with a fast algorithm in contrast to our relaxed upper-bounding formulation). We believe our contributions are the first one to provide a distance which is upper-bounded by a computable divergence.

**Novelty with respect to unbalanced OT:**
We disagree with the statement that our contributions are simple extensions of the existing unbalanced OT theory. The switch from comparing measures to comparing measure spaces requires the use of different proof techniques. The most striking difference is that the associated conic formulations are not on the same cone (UOT considers the cone X x R_+ while we use the 2-D cone R  x R_+). As an example of difference, the proof of definiteness of CGW (appendix C.1.1) requires to take into account  the conic constraints (Eq. (5)), prove its equivalence with a quadratic form of (Eq. (5)), and handle the isometry invariance, which differs radically from UOT. Another illustrative example of difference is when proving the inequality UGW>=CGW: it requires the design of an admissible plan (Eq. (24)) of the latter program. In sharp contrast with the UOT case, proving its admissibility requires taking into account both the conic framework (Eq 25 and computations below) and the quadratic structure of GW (Eq 23) at the same time.

**Use of quadratic f-divergences:**
In addition to these theoretical novelties with respect to unbalanced OT is the use of quadratic divergences (which to the best of our knowledge, is the first apparition of such divergences). Using classical divergences would result in non-homogeneous UGW distances, which would thus degenerate if one considers measures with small or large total mass. Maintaining this 2-homogeneity is crucial and requires new theoretical proof techniques. In the final version, we will further expose these important novelties.

Here is a more precise statement on the degeneracy. Writing $UGW$ and $\tilde{UGW}$ the programs using respectively quadratic and standard f-divergences, we have:
$\lambda^{-2}UGW(\lambda\mu,\lambda\nu) = UGW(\mu,\nu),$
$\lim_{\lambda\rightarrow\infty}\lambda^{-2}\tilde{UGW}(\lambda\mu,\lambda\nu) = 0.$
$\lim_{\lambda\rightarrow\infty}\lambda^{-1}\tilde{UGW}(\lambda\mu,\lambda\nu) = +\infty.$
In other terms, the behaviours differ depending on the scale of the mass, while our formulation remains consistent with respect to rescaling.

**Experiments on real applications:**
We understand the criticism that we did not provide numerical illustrations on real applications. But the point of the paper is to make the case that unbalanced methods make lots of sense for most GW applications. We believe that our work is a key milestone in building efficient and theoretically sound divergences between unbalanced metric measure spaces. Although it is possible to address this by some ad-hoc normalization, we propose the first approach (conic distance) which is theoretically sound and which also comes with a simple to compute upper-bound (relaxed divergence). The purpose of our numerical simulations is rather to illustrate qualitatively the impact of mass relaxation. In the final manuscript, we will show a comparison between KL (already shown) and TV divergence or the methods of [Chapel et al.] to further exemplify the versatility of our approach and the impact of the mass relaxation on the solution.

**Tightness of the upper-bound:**
We believe the upper bound UGW>=CGW is not an equality when the spaces are different, for reasons due to the quadratic structure of the problem. We expect the perspective transform $H_c$ (Eq (10)) not to be tight because the scaling $\theta$ depends on $(x,x’,y,y’,r,r’,s,s’)$, while tightness would require a tensorized dependence in $(x,y,r,s)$ and $(x’,y’,r’,s’)$, in order to be compatible with the structure of GW problem. This being said, even in simple cases (like 2 points spaces), we were not able to find an explicit counter example, and we leave this study for future works.

---

### Decision · Program_Chairs · 2021-01-07
**Final Decision**

**Decision:**

Reject

**Comment:**

This paper present novel formulations to address the problem of unbalanced Gromov. The Conic formulation is very interesting but stays theoretical until optimization algorithms are available. The Unbalanced Gromov is a nice extension of Gromov and comes with relatively efficient solvers. Some very limited numerical experiment show the proposed UGW used between 2D distributions (two moons) and graphs.

The paper had some mixed reviews with reviewers acknowledging the novelty of the approach (albeit an extension similar to unbalanced OT) and of the theoretical results. The detailed a very well written response to the reviewers comment has been appreciated. But all reviewers also noted a lack of numerical experiments outside of the very simple illustrations in the paper. This paper is a very nice contribution to the theory of optimal transport but fails at illustrating its relevance to the ML community.  Despite acknowledging the theoretical contributions of the paper, the  AC recommends a reject but strongly encourages the authors to complete the experimental section with some ML applications or at least proof of concepts (graph classification, domain adaptation, ...).